# Evaluation of the quality of a UAV-based eddy covariance system for measurements of wind field and turbulent flux

Yibo Sun[1,2,3], Shaomin Liu[4], Xinwen Lin[5], Ziwei Xu[4], Bing Geng[6], Bo Liu[1,2,3], Shengnan Ji[1,2,3], Junping Jing[7], Zhiping Zhu[8,9], Bilige Sude[1,2,3], Zhanjun Quan[1,2,3]

[1]State Key Laboratory of Environmental Criteria and Risk Assessment, Chinese Research Academy of Environmental Sciences, Beijing 100012, China.
[2]Institute of Ecology, Chinese Research Academy of Environmental Sciences, Beijing 100012, China.
[3]State Environmental Protection Key Laboratory of Ecological Regional Processes and Functions Assessment, Beijing 100012, China.
[4]State Key Laboratory of Earth Surface Processes and Resource Ecology, Faculty of Geographical Science, Beijing Normal University, Beijing 100875, China.
[5]Collage of Geography and Environment Science, Zhejiang Normal University, Zhejiang 321004, China.
[6]Beijing Academy of Social Sciences, Beijing 100101, China.
[7]National Ocean Technology Center, Tianjin 300112, China.
[8]Kunming General Survey of Natural Resources Center, China Geological Survey, Kunming 650111, China.
[9]Technology Innovation Center for Natural Ecosystem Carbon Sink, Ministry of Natural Resources, Kunming 650100, China.

*Correspondence to*: Yibo Sun (sun.yibo@ craes.org.cn) and Shaoming Liu (smliu@bnu.edu.cn)

**Abstract.** Instrumentation packages of eddy covariance (EC) have been developed for a small unmanned aerial vehicle (UAV) to measure the turbulent fluxes of latent heat (LE), sensible heat (H), and $CO_2$ (Fc) in the atmospheric boundary layer. This study evaluates the measurement performance of this UAV-based EC system. First, the precision ($1\sigma$) of geo-referenced wind measurement was estimated at 0.07 m s$^{-1}$. Then, the effect of calibration parameter and aerodynamic characteristics of the UAV on the quality of the measured wind was examined by conducting a set of calibration flights. The results shown that the calibration improved the quality of measured wind field, and the influence of upwash and leverage effect can be ignored in the wind measurement. Third, for the measurement of turbulent fluxes, the measurement error caused by instrumental noise was estimated at 0.03 µmol m$^{-2}$ s$^{-1}$ for *Fc*, 0.02 W m$^{-2}$ for *H*, and 0.08 W m$^{-2}$ for LE. Fourth, data from the standard operational flights are used to assess the influence of resonance on the measurements and to test the sensitivity of the measurement under the variation ($\pm30$ %) of the calibration parameters around their optimum values. Results shown that the effect of resonance mainly affect the measurement of $CO_2$ (~5 %). The pitch offset angle ($\varepsilon_\theta$) significantly affected the measurement of vertical wind (~30 %) and turbulent fluxes (~15 %). The heading offset angle ($\varepsilon_\psi$) mainly affected the measurement of horizontal wind (~15 %), and other calibration parameters had no significant effect on the measurements. The results lend confidence to use the UAV-based EC system, and suggest future directions for optimization and development of the next generation system.

## 1 Introduction

In environmental, hydrological and climate change sciences, the measurement of surface fluxes at the regional scale (level of several to tens of kilometers) has attracted great interest despite often being considered a gordian knot (Mayer et al., 2022; Chandra et al., 2022). Process-based or remote sensing (RS)-based models are often used to estimate land surface fluxes of matter and energy at continental to global scales with typical spatial resolution from 1-10 km (Hu and Jia, 2015; Mohan et al., 2020; Liu et al., 1999). However, observational data, especially at similar scales to models' estimates, is often lacking, which presents a significant challenge for the validation and evaluation of the surface flux products from these models' estimates (Li et al., 2018; Li et al., 2017). On the ground, in the past decades, extensive eddy-covariance (EC) flux sites with their composed networks and optical-microwave scintillometer (OMS) sites have been built to provide temporally continuous monitoring of surface flux at local (hundreds of meters around the measurement site of ground EC) and path (a distance of a few hundred meters to near 10 kilometers between transmitter and receiver terminal of OMS) scales (Yang et al., 2017; Liu et al., 2018; Zhang et al., 2021; Zheng et al., 2023). Generally speaking, flux from ground measurements need to be scaled up to kilometers-scale to provide comparable spatial surface "relative-truth" flux data for the process- or RS-based models at larger spatial scales (Liu et al., 2016). However, the spatial density of these flux measurements sites is still low compared to the heterogeneity of surface fluxes, which means that major scaling bias may exist in the upscaled flux data (Wang et al., 2016; Li et al., 2021). Therefore, regional-scaled flux measurement techniques need to be developed to complement the ground- and models-based approaches (Vellinga et al., 2010).

Aircraft-based EC flux measurement method, which has been developed for turbulence measurements for more than 40 years (Lenschow et al., 1980; Desjardins et al., 1982), is considered as the optimum method to measure turbulent flux at regional scale (several hundred square kilometers), thus bridging the scale gap between ground and model-derived methods (Gioli et al., 2004; Garman et al., 2006). To date, several types of aircrafts, including manned or unmanned fixed-wing aircrafts, delta-wing aircrafts, and helicopters, have been used for measurements of turbulent flux by equipping them with the EC sensors to measure three-dimensional (3D) wind, air temperature, and gas concentrations at a frequency of 50 Hz (Gioli et al., 2006; Metzger et al., 2012; Thomas et al., 2012; Bange and Roth, 1999). Among them, fixed-wing aircrafts and delta-wing aircrafts are better airborne platforms for EC measurements compared to helicopters due to their tightly coupled structure with the wind sensor and because their flow distortion around the fuselage can be more easily avoided or modeled (Prudden et al., 2018; Garman et al., 2008). A wide range of manned aircrafts has been developed to measure turbulent flux, including single-engine light aircrafts (e.g., Sky Arrow 650, Long-EC, WSMA) (Gioli et al., 2006; Crawford and Dobosy, 1992; Metzger et al., 2012), twin-engine aircrafts (e.g., Twin Otter, NASA CARAFE) (Desjardins et al., 2016; Wolfe et al., 2018) and larger quad-engine utility aircrafts (e.g., NOAA WP-3D) (Khelif et al., 1999). These airborne flux measurements, in combination with ground measurements, provide an excellent opportunity to produce regional-scaled, spatio-temporal continuous surface flux datasets that can improve our understanding of the processes of land-atmosphere interactions in regional and global change (Chen et al., 1999; Liu et al., 1999; Prueger et al., 2005). However, manned aircrafts are expensive to operate and maintain. Aviation

safety and operational regulations require that manned aircrafts must fly above a minimum altitude (400 m above the highest
elevation within 25 km on each side of the center line of the air route) and must avoid hazardous conditions such as icing or
severe turbulence (Elston et al., 2015). The flow distortion induced by the aircraft itself (from the wings, fuselage, and the
propellers) complicates the wind vector measurement from aircraft platform, which means that sophisticated correction
procedures should be applied to compensate for the flow distortion effects (Elston et al., 2015; Williams and Marcotte, 2000;
Drüe and Heinemann, 2013).
In recent years, interesting in unmanned aerial vehicle (UAV) platforms for atmospheric studies have been fast growing,
especially because of their lower construction, operation, and maintenance costs compared with manned platforms. High-
performance fixed-wing UAVs offer a high payload capacity (5-10 kg) and similar endurance (2-3 h) and operating altitude
(up to 3500 m above the sea level) to manned aircrafts, but with much less turbulence disturbance due to their small fuselage
size (Reineman et al., 2013). More importantly, the advancements in small, fast, and powerful sensors and microprocessors
make it possible to use of UAVs for comprehensive atmospheric measurements (Sun et al., 2021a). Several types of UAVs
with different turbulence measurement objectives have been developed and deployed, ranging from small payload capacity
(e.g., 140 g SUMO) to medium (e.g., 1.5 kg $M^2AV$, 1.0 kg MASC) and large (e.g., 6.8 kg Manta, 5.6 kg ScanEagle) (Reuder
et al., 2016; Båserud et al., 2016; Van Den Kroonenberg et al., 2012; Reineman et al., 2013). A comprehensive overview of
the use of these UAVs for turbulence sampling can be found in Elston et al. (2015) and Sun et al. (2021a). For turbulence
measurements, the UAVs were equipped with a commercial or custom multi-hole (5- or 9-hole) probe paired with an integrated
navigation system (INS) to obtain the wind vector. Small and medium UAVs typically could only measure fast 3D wind vector
and air temperature fluctuations for measurements of momentum and sensible heat flux, whereas, large UAVs were equipped
with more types (e.g., radiation, optics, or gas concentration) and more accurate sensors for measurement of a larger range of
meteorological properties including sensible and latent heat fluxes, $CO_2$ flux, radiation fluxes as well as surface properties
(Reineman et al., 2013; Sun et al., 2021a). UAVs equipped with scientific instruments can be deployed in a variety of
application environments and conditions. UAVs offer distinct advantages over manned aircraft in their ability to safely perform
measurements and greatly reduce operational costs especially in low-altitude conditions (below 100 m above the ground level),
which are optimal for measuring turbulent flux (Witte et al., 2017). Anderson and Gaston (2013) predict that UAVs will
revolutionize the spatial data collection in ecology and meteorology.
EC method is a well-developed technology for directly measuring vertical turbulent flux (flux of sensible heat, latent heat
and $CO_2$) within the atmospheric boundary layers (ABL) (Peltola et al., 2021). It requires accurate time (for ground tower) or
spatial (for mobile platform) series of both the transported scalar quantity and the transporting turbulent wind. Each should be
measured at sufficient frequency to resolve the flux contribution from small eddies (Vellinga et al., 2013). The measurement
of the geo-referenced 3D wind vector, which is the prerequisite for EC measurements, is challenging for airborne platform.
Airborne measurement of geo-referenced 3D wind is the vector sum between the aircraft velocity relative to the earth (inertial
velocity) and the velocity relative to the air (relative wind vector, or true airspeed). Therefore, accurate measurements of the
relative wind as well as the motion and attitude of the platform are essential to accurately measure the geo-referenced wind
vector and turbulent flux (Metzger et al., 2011). Garman et al. (2006) estimated the measurement precision ($1\sigma$) of the vertical
wind measurements of a commercial 9-hole turbulence probe (known as "Best Air Turbulence Probe", often abbreviated as
the "BAT Probe") to be 0.03 m s$^{-1}$ by combining the precision of the BAT Probe and the integrated navigation device. The
BAT Probe is widely used on manned fixed-wing aircrafts, such as Sky Arrow 650 ERA (Environmental research aircraft),
Beechcraft Duchess, and Diamond DA42, for turbulent flux measurement (Gioli et al., 2006; Garman et al., 2008; Sayres et
al., 2017). A light delta-wing EC flux measurement aircraft developed by Metzger et al. (2011) reported a $1\sigma$ precision of
wind measurement of 0.09 m s$^{-1}$ for horizontal wind and 0.04 m s$^{-1}$ for vertical wind using a specially customized five-hole
probe (5HP). On this basis, in combination with a commercial infrared gas analyzer, the $1\sigma$ precision of flux measurement
was 0.003 m s$^{-1}$ for friction velocity, 0.9 W m$^{-2}$ for sensible heat flux, and 0.5 W m$^{-2}$ for latent heat flux (Metzger et al., 2012).
The EC flux measurement from a UAV platform can now be achieved with a similar reliability to a manned platform. The
Manta and ScanEagle UAV-based EC measurements developed by Reineman et al. (2013) achieved precise wind
measurements (0.05 m s$^{-1}$ for horizontal and 0.02 m s$^{-1}$ for vertical wind) using a custom nine-hole probe and a commercial
high precision  integrated navigation system (INS), at a lower price and lighter weight than the commercial BAT probe.
However, the onboard instrument packages for Manta and ScanEagle UAV are independent of each other in their
measurements of turbulent and radiation flux, and the $CO_2$ flux measurement is lacking.
Inspired by these studies, Sun et al. (2021a) used a high-performance fuel-powered vertical take-off and landing (VTOL),
fixed-wing platform to integrate the scientific payloads for EC and radiation measurements to obtain a comprehensive
measurement of turbulent and radiation flux using an UAV. This UAV-based EC system measured turbulent fluxes of sensible
heat, latent heat, and $CO_2$, as well as radiation including net radiation and upward- and downward-looking photosynthetically
active radiation (PAR). This system was successfully tested in the Inner Mongolia of China and applied to measure the regional
sensible and latent heat fluxes in the Yancheng coastal wetland in Jiangsu, China (Sun et al., 2021a; 2021b). During these field
studies, the UAV-based EC measurements achieved a near consistent observational result compared with ground EC
measurements (Sun et al., 2021b). However, some shortcomings in the developed UAV-based EC system were also identified.
In particular, the noise effects from the engine and propeller were not fully isolated, resulting in high frequency noise in the
measured scalars (air temperature, $H_2O$, and $CO_2$ concentration). This UAV-based EC system is being continuously improved
(in Section 2.1) based on field measurements. However, there is no quantitative evaluation of the measurement precision of
the wind field and turbulent flux as well as of the influence of the resonance noise from the UAV operation yet. Previous work
using ground EC as a benchmark to assess the measurement performance of the UAV-based EC system has been disputed,
due to difference in EC sensors, platforms, measurement height, and source areas (i.e., footprint), as well as the influence of
surface heterogeneity, flux divergence, inversion layer and the stochastic nature of turbulence (Sun et al., 2021b; Wolfe et al.,
2018; Hannun et al., 2020).
This study attempts to evaluate the performance of the UAV-based EC system developed by Sun et al. (2021a) in the
measurement of wind field and turbulent flux. For these purposes, data from two field measurement campaigns, including a
set of calibration flights and some standard operation flights, were used in this study. First, the current study investigated the
quality of the measurement of geo-referenced wind vector including measurement error ($1\sigma$) and the improvements for wind
measurement after system calibration. Second, using the measured data from standard operation flights, flux measurement
error related to instrumental noise was estimated with a method proposed by Billesbach (2011). Errors propagated through the
correction terms [i.e., Webb-Pearman-Leuning (WPL) correction for latent heat and $CO_2$ flux] were also included in our
analysis (Webb et al., 1980; Kowalski et al., 2021). Then, the impacts of resonance noise on the measured scalar variance and
the flux covariance were also estimated by comparing the real (co)spectra curve with the theoretical reference curve from
Massman and Clement (2005). Lastly, the sensitivity of the measured geo-referenced wind vector and turbulent flux to the
errors in the calibration parameters (determined by the calibration flight) were assessed by adding an error of ±30 % to their
optimum calibration value.

## 2 Materials and Methods

### 2.1 The UAV-based EC system

The UAV platform used for EC measurement is a high-performance, fuel-powered VTOL, fixed-wing UAV, which has
minimal requirements for the takeoff location and offers a high payload capacity of up to 10 kg. The UAV has a wing-span of
3.7 m, a fuselage length of 2.85 m, and a maximum take-off weight of 60 kg. The UAV engine is mounted in a pusher
configuration, allowing for the turbulence probe to be installed directly on the nose of the UAV, minimizing or eliminating
airflow contamination due to upwash and sidewash generated by the wings (Crawford et al., 1996). Control of the UAV is
totally autonomous, and the pilots have the option to enable manual and semi-manual control in emergency conditions. The
UAV has a cruise flight speed of 28 to 31 m s$^{-1}$ with an endurance of almost 3 h, and it has a flight ceiling of up to 3800 m
above the sea level. Detailed information on this UAV could be found in Sun et al. (2021a).
The flux payloads of the UAV-based EC system include a precision-engineered 5-hole pressure probe (5HP) for
measurement of the true airspeed and the attack ($\alpha$) and sideslip ($\beta$) angles of incoming flow relative to the UAV, a dual-
antenna integrated navigation system (INS) for high accuracy measurement of UAV ground speed and attitude, an open path
infrared gas analyzer (IRGA) for recording the atmospheric densities of $CO_2$ and water vapor, a fast temperature sensor for
measurement of the fast temperature fluctuations, and a slow-response temperature probe for providing a mean air temperature
reference. The sample rate is 50 Hz except for the slow-response temperature probe (1 Hz), yielding a turbulence horizontal
resolution of approximately 1.2 m at a cruising speed of 30 m s$^{-1}$. The system was improved according to deficiencies identified
after several field measurements with the following adjustments: 1) a laser distance measurement unit was mounted for
measuring the distance between the UAV and the ground level, 2) the platinum resistance thermometer was replaced by a
thermocouple (Omega T-type COCO-003; $\varnothing$0.075 mm) for improving the resistance of the high-frequency temperature
measurements to vibration noise from the engine, 3) the vibration isolator structure of the IRGA was improved, and 4) the
original datalogger (CR1000X, Campbell, USA) was replaced with a lighter one (CR6, Campbell, USA). All the digital and
analog signals from the sensor modules are stored and synchronized by the on-board datalogger, and the on-board scientific
payloads are designed to be isolated from the electronic components of the UAV to ensure that any problems occurring would
not jeopardize the safety of the UAV (Sun et al., 2021a).
In the present study, to estimate the measurement precision of the geo-referenced wind and turbulent flux, the sensor
modules and their $1\sigma$ precision of the measured variables related to EC measurement were used, as presented in Table 1. For
the 5HP, the $1\sigma$ measurement precision was acquired from the wind tunnel test after wind tunnel calibration (Sun et al., 2021a).
**Table 1**: Summary of the sensor modules, measured variables, and their measurement precision used to determine the geo-
referenced wind velocity and turbulent flux.

| Sensor (Module, company, country) | Variables | Precision ($1\sigma$) |
|---|---|---|
| GNSS/INS (BD992-INS, Trimble, USA) | Roll, Pitch, Heading | 0.1° |
| | Horizontal velocity | 0.007 m s$^{-1}$ |
| | Vertical velocity | 0.02 m s$^{-1}$ |
| 5HP (ADP-55, Simtec AG, Switzerland) | Attack angle | 0.02°[#] |
| | Sideslip angle | 0.04°[#] |
| | True airspeed | 0.05 m s$^{-1}$[#] |
| | Static pressure | 1.1 hPa |
| | Dynamic pressure | 0.003 hPa |
| IRGA (EC150, Campbell, USA) | $CO_2$ density | 0.2 mg m$^{-3}$ |
| | $H_2O$ density | 0.004 g m$^{-3}$ |
| Thermistor (100K6A1IA, Campbell, USA) | Temperature (slow) | 0.2 ℃ |
| Thermocouple (T-type COCO-003, Omega, USA) | Temperature (fast) | 0.5 ℃ |

[#] Results from the wind tunnel test.

## 2.2 Field campaign

### 2.2.1 In-flight calibration campaign

In order to calibrate the wind measurement component of the UAV-based EC system, an in-flight calibration campaign was
carried out on 4 September 2022 at the Caofeidian Shoal Harbor in the Bohai Sea of northern China. The average water depth
of this area is approximately 0-5 m, with a maximum water depth of 22 m. At low tide, a large area of the tidal flat is exposed;
while at high tide, only the barrier islands are visible (Xu et al., 2021). The assumptions for calibration flight include 1) low
turbulent transport condition (i.e., no disturbance), 2) a constant mean horizontal wind, and 3) mean vertical wind near zero
(Drüe and Heinemann, 2013; Vellinga et al., 2013; Van Den Kroonenberg et al., 2008). This allows identical wind components
for several consecutive straights in opposite or vertical flight directions. These assumptions are usually well satisfied above
the ABL or under stable atmospheric conditions (Drüe and Heinemann, 2013). Over the sea surface, due to its uniform and
cool surface property, the turbulence fluctuations are weaker than that over the land surface (Mathez and Smerdon, 2018),
making where a more ideal environment to conduct calibration flight.
The in-flight calibration campaign included three flight maneuvers, including a 'box' maneuver, 'racetrack' maneuver, and
'acceleration-deceleration' maneuver. The trajectory of the calibration flight is shown in Figure 1, with different color
corresponding to different flight maneuver. The calibration flight was executed between 7:28-7:48 a.m. (China Standard Time,
CST) to coincide with the ebb tide stage. During this time, the average water depth was approximately 1.1 m, and the average
flight altitude was 400 m ($\sigma = \pm 0.78$ m) above the sea level. Considering the uniform and cool underlying surface and the
stable atmospheric conditions in the early morning, we assume no disturbance from underlying surface was present during the
calibration flight.

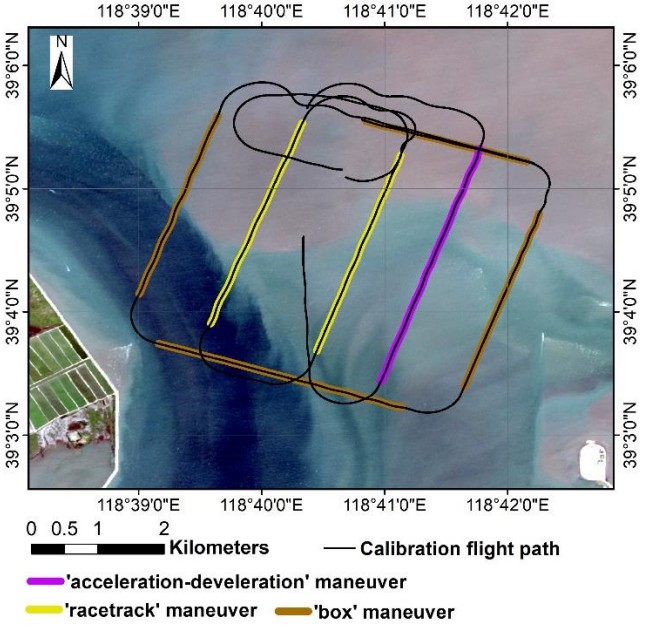


**Figure 1. Flight trajectory of the calibration campaign on 4 September 2022 at the Caofeidian Shoal Harbor in the Bohai Sea of northern China. The land surface image is from Sentinel-2A satellite image with true color combination acquired on 1 September 2022.**

In this study, the 'box' maneuver is used to determine the mounting misalignment angle in the heading ($\epsilon_\psi$) and pitch ($\epsilon_\theta$)
between the 5HP and the center of gravity (CG) of the UAV. The flight path is a box in which the four straight legs are flown
at constant cruising speed, flight altitude, and heading for 2 minutes. The 'racetrack' maneuver is used to evaluate the quality
of the calibration parameters acquired from the previous 'box' maneuver. The flight path consists of two parallel straight flight
tracks connected by two $180^\circ$ turns. Each straight flight section lasts 2 minutes at constant speed and flight altitude. Lastly,
the 'acceleration-deceleration' maneuver is used to check the influence of lift-induced upwash from the wing to the measured
attack angle by the 5HP. During this maneuver, the aircraft is kept straight and level at constant pressure altitude. When
beginning this maneuver, the aircraft accelerates to its maximum airspeed (35 m s$^{-1}$). Then, the airspeed reduces gradually to
near its minimum airspeed (25 m s$^{-1}$) and back up to its maximum airspeed. The pressure-altitude of the aircraft is maintained
throughout this maneuver, and the entire maneuver lasts one minute. This maneuver creates a series continuous changed pitch
($\theta$) and attack ($\alpha$) angle. A relationship between the measured incident flow attack angles ($\alpha$) by the 5HP and the measured
pitch angle by the INS of close to 1:1, indicates that the effect from the fuselage-induced flow distortion on the wind
measurements is negligible (Garman et al., 2006).

## 2.2.2 Standard operation flight campaign

The reliability of the EC measurement from UAV is susceptible to several factors, mainly including instrumental noise,
resonance noise, and the quality of the calibration parameter. In order to evaluate the flux measurement error related to
instrumental noise, the effects of resonance on the measured scalar and to investigate the sensitivity of the measured geo-
referenced wind and turbulent flux to uncertainty in the calibration parameter, we used data from 7 flights in the Dagang
district in Tianjin, China between 8 and 16 August 2022. This area is located on the west coast of the Bohai Sea and is a coastal
alluvial plain with altitudes between 1-3 m (Chen et al., 2017). The flight path, shown in Figure 2, includes three parallel
transect lines of approximately 4 km in length each and at 1-2 km intervals. All flights occurred during the daytime, and were
performed in the same trajectory at low altitude about 90 m above sea level. The flight area covered three different underlying
surfaces: land, coastal zone, and water surfaces, that can represent typical flux intensity characteristics for different surface
conditions.

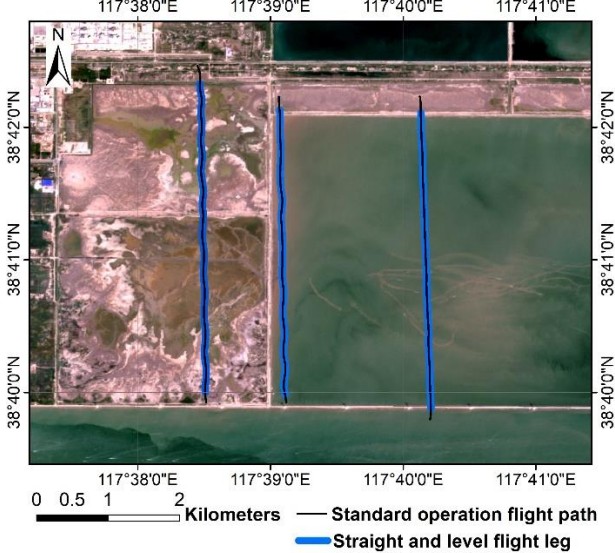


**Figure 2. Flight trajectory of the standard operation flight campaign, 8-16 August 2022, at Dagang district, Tianjin, China. The land**
**surface image is from Sentinel-2A satellite image with true color combination acquired on 27 August 2022.**
During the operation flight campaign, the atmospheric stability conditions changed from stable (Monin-Obukhov stability
parameter, $z/L = 1.93$) to very unstable ($z/L = -10.28$) as measured by the UAV, where $z$ is the flight height above the
ground level, $L$ is the Obukhov length. The stable condition mostly occurred on flight path located over the sea surface, while
the unstable condition mostly occurred on flight path located over the land surface. These flight data provide various
measurement conditions for us to evaluate the performance of the UAV-based EC system.

## 2.3 Data processing

The raw data collected with the on-board datalogger (CR6, Campbell, USA) is subsequently saved in Network Common Data
Form (netCDF) format. It includes dynamic and static pressure, attack, and sideslip angle of incoming flow; slow (1 Hz) and
fast (50 Hz) air temperature; mass concentration of $H_2O$ and $CO_2$; as well as the full navigation data (including 3D location,
ground speed, angular velocity, and attitude, etc.) of the UAV. The subsequent data processing includes three basic processing
stages in order to calculate flux data from raw measured data.
In the first stage, a moving average filter was used to detect outliers in each variable. Detected outliers were removed and
replaced by values obtained by linear interpolation. Outliers tend to be rare. However, if outliers constitute more than 20 % of
the data points, the corresponding flight data should be discarded. The cleaned raw data was then used to calculate the geo-
referenced wind vector, (co)spectra, and turbulent fluxes.
In the second stage, geo-referenced 3D wind vector is calculated. The full form of the equations of motion for calculating
the geo-referenced wind vector by the UAV-based EC system is described in detailed in Supplement Part A. From the aircraft
platform, geo-referenced wind vector is measured in two independent reference coordinate systems: the relative true airspeed
($\widehat{U}_a$) measurement in the aircraft coordinate system and the ground speed of the aircraft ($U_p$) in the geo-referenced coordinate
system. The geo-referenced wind ($U$) is the vector sum of the relative true airspeed ($\widehat{U}_a$), the UAV's motion ($U_p$) and the
tangential velocity due to the rotational motion of the aircraft ("lever arm" effect), which is described in Eq. (S2). In this stage,
the acquired calibration parameters ($\epsilon_\psi$ and $\epsilon_\theta$) from the calibration flight are substituted into the Eq. (S8) to correct the
mounting angle offset errors between the 5HP and the CG of the UAV. The final equations for geo-referenced wind vector
calculation (Eqs. S15 to S17) revealed that the lever arm effects due to the spatial separation between the tip of the wind probe
and the CG of UAV can influence the wind measurements. Typically, the separation distance ($L$) is small, and the influence
of the lever arm effects can be ignored when the $L$ less than about 10 m (Lenschow, 1986). In the current UAV-based EC
system, the displacements of the 5HP tip with respect to the CG of the UAV along the three axes of UAB body coordinate are:
$x^b = 1.459$ m, $y^b = 0$ m, and $x^b = 0.173$ m (in Supplement Part A). Therefore, in practice, the influence of leverage effects
in geo-referenced wind calculation was also ignored in this study. This was confirmed by assessing the difference in the geo-
referenced wind vector with and without the leverage effect correction term (in Section 3.1).
In the final stage, based on the EC technology and spatial averaging, the turbulent flux is calculated using the covariances
of vertical wind ($w$) with air temperature ($T_a$) for sensible heat flux ($H$), with water vapor density ($q$) for latent heat flux (LE),
and with $CO_2$ density ($c$) for $CO_2$ flux ($F_c$), and with the necessary correction (Webb et al., 1980). The time lag due to the
separation between the 5HP tip, the adjacent temperature probe, and the open-path gas analysis did not need to be corrected
because the time delay was less than 1 second at the cruise airspeed of 30 m s$^{-1}$ and sensor separation less than 20 cm. Only
the measurement data from the straight-line portion of the flight path can be used in flux calculation. Detailed calculation
procedure and formulas of $H$, LE, and $F_c$ used by the present UAV-based EC system are provided in Supplement Part B,
including spatially averaging, coordinate rotation, and necessary correction (i.e., WPL correction for LE and $F_c$).
One important aspect for airborne EC measurement is the definition of a proper spatial averaging length to calculate
turbulent flux (Sun et al., 2018). Such spatial averaging length depends on the flying altitude, surface characteristics, and
atmospheric stability, and could be determined using Ogive analysis (Gioli et al., 2004; Kirby et al., 2008). In this study, the
entire measured data of each straight and level flight leg (each with length about 4 km) from the standard operational flight
campaign was used to calculate turbulent flux, regardless of the uncertainty in fluxes associated with spatial averaging.
**2.4 Evaluation scheme**
**2.4.1 Wind measurement evaluation**
The key to successful aircraft EC measurements lies in the translation of accurately measured, aircraft-orientated, wind vector
to geo-referenced orthogonal wind vector (Thomas et al., 2012). Determining the geo-referenced wind vector requires a
sequence of thermodynamic and trigonometric equations (Metzger et al., 2012), these equations propagate various sources of
error to the measured geo-referenced wind vector. To estimate the measurement errors in the geo-referenced wind vector, we
used the linearized Taylor series expansions of Eqs. (S15) to (S17) derived by Enriquez and Friehe (1995) (in Supplement Part
A) to determine the sensitivities of each of the geo-referenced wind vector components with respect to the relevant variables.
Then, these sensitivity terms can be combined to compute the overall measurement error ($1\sigma$) in the geo-referenced 3D wind
vector (Eqs. S21 to S23 in Supplement Part A).
The above wind measurement error analysis gives the nominal measurement precision of the geo-referenced wind, but does
not consider the influence of environmental changes. Following the methods of Lenschow and Sun (2007), we assess whether
the accuracy of wind measurements from the UAV in satisfying the minimum signal level needed for resolving the mesoscale
variations of the three wind components in the encountered atmospheric conditions. Firstly, the minimum required signal level
for measurement of vertical air speed ($\omega$) under the encountered atmospheric conditions could be estimated as (Lenschow and
Sun, 2007):
$\frac{\partial w}{\partial t} < 0.2\sqrt{2}\sigma_w 2\pi k U_a$ (1)
with the true airspeed ($U_a$) set to mean cruise speed 30 m s$^{-1}$, the peak signal magnitude ($\sigma_w$) of the power spectra, and the
corresponding wavenumber ($k$) (Thomas et al., 2012). The measurement error of the system in the vertical wind component
can be calculated as (Lenschow and Sun, 2007):
$\frac{\partial w}{\partial t} \cong \Theta \frac{\partial U_a}{\partial t} + U_a \frac{\partial \Theta}{\partial t} + \frac{\partial w_{UAV}}{\partial t}$ (2)
with $\Theta = \alpha - \theta$, where $\alpha$ is the attack angle, $\theta$ is the pitch angle, $w_{UAV}$ is the UAV's vertical velocity. According to Lenschow
and Sun (2007), the signal level and mesoscale fluctuation of horizontal wind components ($u$ and $v$) are considerably larger
than that of vertical wind, so the accuracy criteria are not nearly as stringent. The measurement error of the horizontal wind
component could be calculated as (Lenschow and Sun, 2007):
$$\frac{\partial u}{\partial t} \cong -\frac{\partial U_a}{\partial t} + \frac{\partial u_{UAV}}{\partial t} \qquad (3)$$
$$\frac{\partial v}{\partial t} \cong \Psi \frac{\partial U_a}{\partial t} + v_{tas}\frac{\partial \Psi}{\partial t} + \frac{\partial v_{UAV}}{\partial t} \qquad (4)$$
and,
$$\Psi \equiv \psi' + \beta \qquad (5)$$
where $u_{UAV}$, $v_{UAV}$ are the UAV's horizontal velocity measured from INS, $\psi'$ is the departure of the measured true heading
from the average true heading, and $\beta$ is the sideslip angle of airflow. If the measurement error of the 3D wind vector from Eqs.
(2) to (4) is smaller than the required minimum signal level of the vertical and horizontal wind components, it can be confirmed
that the measurement accuracy of the geo-referenced 3D wind vector from UAV is sufficient to resolve the mesoscale
variations of the three wind components in the encountered atmospheric conditions.
In addition, accurate measurements of geo-referenced wind vector typically not only depend on the measurement precision
of the sensors (i.e., 5HP and INS), but also related to the quality of the calibration parameters and the geometry structure of
the UAV EC system (i.e., flow distortion and leverage effect). For evaluation of the effect of the latter two aspects, a calibration
flight campaign (Section 2.2.1) was performed to determine the calibration parameter ($\epsilon_\psi, \epsilon_\theta$), check its quality, as well as to
ascertain the effects of the lever arm and up-wash by the wings. The methods for acquiring the calibration parameter were
given by Vellinga et al. (2013) and Sun et al. (2021a), and the results are reported in Supplement Part C (Figs. S2 and S3).
During the in-flight calibration campaign, a 'racetrack' maneuver was performed to check the quality of the calibration
parameters determined from the 'box' flight maneuver. The initial ($\epsilon_\psi = 0°, \epsilon_\theta = 0°$) and calibrated ($\epsilon_\theta = -0.183°, \epsilon_\psi = 2°$,
in Supplement Part C) set of parameters were used to calculate the geo-referenced wind vector. By comparing the mean and
standard deviation of the horizontal and vertical wind vector between the initial and calibrated set, the quality of the geo-
referenced wind vector measurement in real environment conditions can be verified.
The relative wind vector ($\widehat{U}_a$) measured by the aircraft is susceptible to flow distortion because the airplane must distort the
flow to generate lift and thrust, and the aircraft's propellers, fuselage, and wings are the main sources of flow distortion as
flow barriers (Metzger et al., 2011). For fixed-wing aircrafts, the wind probe mounted on the nose of the UAV and extended
as far forward of the fuselage as possible could avoid significant influence from flow distortion from the fuselage and propellers.
Nevertheless, effects from the induced upwash by the wings can also significantly influence the correspondence between
measured and free-stream flow variables (Garman et al., 2008). The induced upwash by the wings modifies the local angle of
attack, causing the measured attack angle ($\alpha$) to be larger than the free-stream attack angle ($\alpha_\infty$). The magnitude of upwash
influence generally increases with airplane size and airspeed, typically ranging from 0.5 to 2.5 m s$^{-1}$ as reported by the manned
fixed-wing aircraft (Garman et al., 2008). Therefore, for wind measurements by manned fixed-wing aircrafts, the upwash
effects must be corrected (Garman et al., 2008; Kalogiros and Wang, 2002). However, wind measurements using a multi-hole
probe on the UAV seldom need this correction due to the fuselage size and because the airspeed is very low compared to a
manned aircraft.
In order to access whether the lift-induced upwash could be safety ignored by the current UAV-based EC system, the
'acceleration-deceleration' flight maneuver was performed. According to Crawford et al. (1996), the pitch angle ($\theta$) by the
INS instrument can be utilized as an estimate of the free-stream attack angle ($\alpha_\infty$) if the aircraft's vertical velocity is zero,
since it is unaffected by lift-induced upwash and varies directly with $\alpha_\infty$ when the ambient vertical wind is zero. Under ideal
conditions (zero aircraft vertical velocity and zero ambient vertical wind), the approximation relationship of $\theta \cong \alpha_\infty$ is valid
when $\theta < 6°$ (Crawford et al., 1996; Vellinga et al., 2013). Departures from the 1:1 relationship can be caused by airflow
distortion around the airplane behind the 5HP. The 'acceleration-deceleration' maneuver produced various pitch and attack
angles measured under various airspeeds, which allowed a direct comparison between the pitch angle ($\theta$) and the attack angle
($\alpha$). If the slope between $\alpha$ and $\theta$ is close to unity, it indicates that the influence of lift-induced upwash can be ignored;
otherwise, its influence should be corrected using upwash models (Garman et al., 2006). Meanwhile, the influence of leverage
effects was also evaluated based on the measurement data from the 'acceleration-deceleration' maneuver by considering or
ignoring the leverage effect correction term in Eqs. (S15) to (S17).
**2.4.2 Flux measurement error caused by instrumental noise**
Errors or uncertainties in EC measurements can be systematic or random. Measurement from UAV, they can be attributed to
several sources, mainly including instrumental noise, data handing, atmospheric conditions, insufficient flux calculation length,
and bumpy flight environment (Mahrt, 1998; Finkelstein and Sims, 2001; Mauder et al., 2013). Knowledge of the measurement
precision is of great importance for interpretation of EC measurements especially when detecting small fluxes in terms of
turbulent exchange or signal-to-noise (SNR) of the instrumentation. Determination of the flux measurement error from
instrument noise is very useful, as it is related not only to the system performance, but also to the minimum resolvable
capability for the flux to be measured. In the current study, uncertainty related to instrumental noise (listed in Table 1) was
estimated with a directly method proposed by Billesbach (2011). This method can be called as "random shuffle" (denoted as
the RS) method and was "designed to only be sensitive to random instrument noise". According to Billesbach (2011), the
uncertainty of the flux covariance can be expressed as:
$\sigma_{\overline{w'x'}} = \frac{1}{N} \sum_{i,j=1}^{N} w'(t_i) x'(t_j)$                                                      (6)
where $x$ is the target entity of the covariance, $N$ is the number of measurements contained in the block averaging period, $j \in$
$[1 \dots N]$ but the values are in the random order. The idea behind the RS method was that the randomly shuffled will remove
the covariance between biophysical (source/sink) and transport mechanisms, leaving only the random "accidental" correlations
attributed mostly to instrument noise (Billesbach, 2011). It means that the shuffled component $x$ makes it uncorrelated in
time/space and decorrelates $x$ from $w$, resulting in two independent variables (i.e., $\overline{w'x'} \sim 0$), and any residual value of the
covariance is attributed to random instrument noise. In addition to the basic assumptions made in EC flux calculation, one
additional assumption in RS method is that the block averaging period should be sufficiently long to accurately represent the
lowest significant frequencies contributing to the covariance which could be verified by forming Ogive plots of the covariance
(Billesbach, 2011).
In this study, in order to obtain robust estimates of the instrumental noise, $\sigma_{\overline{w'x'}}$ was repeatedly calculated 20 times for every
straight and level flight leg in operation flight (Fig. 2), and the mean of the absolute values of these 20 replicated estimates for
$\sigma_{\overline{w'x'}}$ were used to estimate the random uncertainty related to instrumental noise. According to Rannik et al. (2016), RS method
tends to overestimate the covariance uncertainty due to instrumental noise only. Then, the uncertainty in the flux covariance
of sensible heat ($\sigma_{\overline{w'T'}}$), latent heat ($\sigma_{\overline{w'\rho'_v}}$), and $CO_2$ ($\sigma_{\overline{w'\rho'_c}}$) were estimated using RS method, respectively.
It should be noted that the measurement error of EC flux measurement is influenced not only by the uncertainty in the raw
covariance but also by the propagated errors form correction terms (i.e., WPL correction) or any lens contamination (Serrano-
Ortiz et al., 2008). For EC measurement from our UAV, the signal quality of the IRGA is checked before each flight
measurement to ensure that the measurement of gas concentration is not affected by lens contamination. The final uncertainty
of flux measurement was evaluated using the partial derivatives of the full flux calculation equation with respect to their flux
value derived by Liu et al. (2006) (Eqs. S29 to S31 in Supplement Part B). These equations (Eqs. S29 to S31) ignored the
perturbations terms from the errors in the individual scalar (i.e., $\rho_v$, $\rho_c$, $T$) which were proved negligible small (Serrano-Ortiz
et al., 2008). At last, after several repetitive calculation of the Eq. (6), their averaged value then be combined to Eqs. (S29) to
(S31) for estimating the final flux measurement error due to instrumental noise.

### 2.4.3 Resonance effects

Previous work found that the measurement of the atmospheric scalars (e.g., air temperature, $H_2O$, and $CO_2$ concentration) is
susceptible to resonance effects caused by the operation of the engine and propeller (Sun et al., 2021b). In order to further
reduce the resonance effects, the vibration damping structure of the developed UAV-based EC system was further optimized.
The reference (co)spectra curve of Massman and Clement (2005) was used to quantify the influence of the resonance effects
remaining after vibration isolation optimization. Massman and Clement (2005) gave the generalization mathematical
expression of the models of spectra and co-spectra:
$$Co(f) = A_0 \frac{1/f_x}{[1 + m(f/f_x)^{2\mu}]^{\frac{1}{2\mu}(\frac{m+1}{m})}} \tag{6}$$
where $f$ is frequency (Hz), $f_x$ is the frequency at which $fCo(f)$ reaches its maximum value, $A_0$ is a normalization parameter,
$m$ is the (inertial subrange) slope parameter, and $\mu$ is the broadness parameter. To describe co-spectra, $m$ should be 3/4; to
describe spectra, $m$ should be 3/2. According to Massman and Clement (2005), $\mu = 7/6$ under stable atmospheric condition
and $\mu = 1/2$ under unstable atmospheric condition. Fast Fourier transform (FFT) method was used to calculate the spectra
and co-spectra of the measured turbulent variables. Before calculating the turbulence (co)spectra, condition of the raw
turbulence data was performed, including a linear detrend and tapering using the Hamming window to reduce the spectral
leakage (sharp edge) according to Kaimal et al. (1989).
According to Sun et al. (2021b), the noise influence from resonance mainly appears in the high frequency domain. According
to the feature of spectral curve, the frequency range of the noise region was artificially designated to $f > 8$ Hz for air
temperature, $f = 1 \sim 5$ Hz for water vapor, and $f = 1 \sim 8$ Hz for $CO_2$. The normalized spectra and co-spectra curve were
adopted and the area difference of the designated frequency range beneath the (co)spectra curve between the measured and
reference (co)spectra curve was calculated to quantify the influence of resonance noise in the variance and flux covariance of
the measurement atmosphere scalars. An example is shown in Figure 3, and also shown is the reference (co)spectra curve of
Massman and Clement (2005), with the (co)spectral maximum at $f_x = 0.1$. The red region in Fig. 3 represents the impact
extent of the resonance noise in the variance (Figs. 3a to 3c) and flux covariance (Figs. 3d to 3f) of the measured scalars. The
systematic noise deviation in the fluxes of sensible, latent heat and $CO_2$ could be derived relative to the entire frequency range.

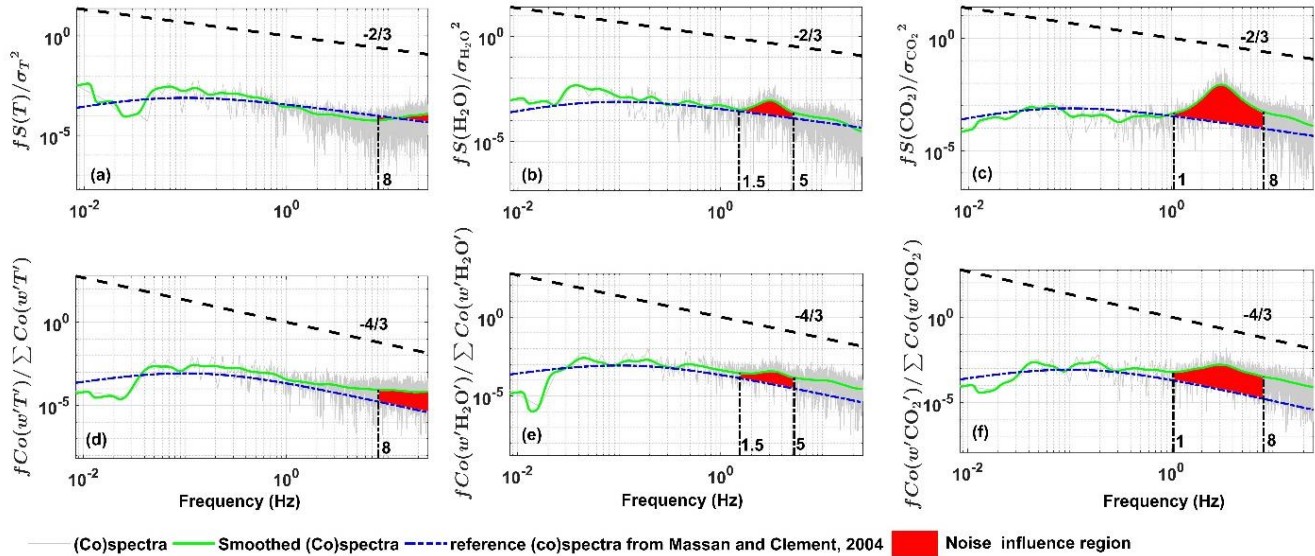


**Figure 3. The influence of resonance noise on the spectra (top row) and co-spectra (bottom row) curve of the measured scalars based**
**on the measurement from the standard operation flight campaign on 8 August 2022 at Dagang district, Tianjin, China. The red**
**region is the area difference of the designated frequency range (vertical black dashed-dotted line) beneath the (co)spectral curve**
**between the measured and reference (co)spectral curve.**
**2.4.4 Sensitivity analysis**
To understand the relevance of the calibration parameters for the measurement of geo-referenced wind vector and turbulent
flux, two sensitivity tests were conducted. The magnitude of the perturbation in the wind vector and turbulent flux was
investigated as a function of the uncertainties in the four calibration parameters, including three mounting misalignment angles
$(\epsilon_\psi, \epsilon_\theta, \epsilon_\phi)$ between the 5HP and the CG of the UAV and one temperature recover factor ($\epsilon_r = 0.82$) used to calculate the
ambient temperature (Eq. 3 in Sun et al. 2021a).
First, the sensitivity of the geo-referenced 3D wind vector and turbulent flux to the uncertainties of the individual parameter
was investigated. The geo-referenced 3D wind vector and turbulent flux was calculated based on the straight leg (about 4 km)
of the standard operational flight by adding an error of $\pm30$ % to the optimum value of each calibration parameter alternately;
except for $\epsilon_\phi$, for which the typical range of $\pm0.9°$ was taken for sensitivity analysis (Vellinga et al., 2013).
Then, in order to test the overall interaction between the parameters, a second sensitivity test was performed to calculate the
geo-referenced 3D wind vector and turbulent fluxes by adding $\pm30$ % error to all calibration parameters simultaneously.
Lastly, their relative errors (*RE*) with respect to the original value were calculated to evaluate the perturbation of the wind
vector and turbulent flux under the variation of each calibration parameter as well as under simultaneous variation of all
calibration parameters. During the sensitivity analysis, the calculated geo-referenced wind and turbulent flux results whose
absolute value was less than their least resolvable magnitude were filtered out to avoid the influence of the errors contained in
the measurements themselves on the results.
**2.4.5 Relative error**
In this study, relative error (*RE*) was used to evaluate the influence of different factors on the measurements of geo-referenced
wind vector and turbulent flux by the UAV-based EC system. It is defined as:
$RE = \frac{|x_0| - |x|}{|x|} \times 100 \%$  (7)
where '| |' means the absolute value, $x$ is the 'true' value, $x_0$ is the influenced value. *RE* > 0 means the exerted influence will
cause the measurement value to be larger than 'true' value and vice versa.
**3 Results**
**3.1 Wind measurement evaluation**
Evaluation of the wind measurement performance from the UAV-based EC system includes three contents: (1) measurement
precision and its ability to resolve the mesoscale variations of the wind, (2) checking the quality of the acquired calibration
parameters, and (3) checking whether the measured wind vector is affected by upwash flow and leverage effects.
First, according to the equations described in Supplement Part A (Eqs. S18 to S23), the measurement precision for horizontal
wind components is a function of true airspeed and true heading, while, the measurement precision for vertical wind component
is largely decided by the true airspeed. The typical values of true airspeed ranging from 25 m s$^{-1}$ to 35 m s$^{-1}$ (interval of 1 m s$^{-1}$
$^1$) and the true heading values ranging from 0° to 180° (interval of 30°) were used in the evaluation of wind measurement
error. Then, the measurement precision (1$\sigma$) of the geo-reference 3D wind vector from aircraft was estimated using the
measurement precision of the related parameters from Table 1. The results are shown in Figure 4 for the measurement precision
of horizontal wind ($\sigma_u$ and $\sigma_v$ in Figs. 4a and 4b) and vertical wind ($\sigma_w$ in Fig 4c), respectively. The typical values of the
measurement precision are ranging from 0.05 m s$^{-1}$ to 0.07 m s$^{-1}$ for horizontal wind component $u$, ranging from 0.02 m s$^{-1}$ to
0.08 m s$^{-1}$ for horizontal wind component $v$, and ranging from 0.05 m s$^{-1}$ to 0.07 m s$^{-1}$ for vertical wind component $w$. When
the flight direction is towards due east or due west, the horizontal wind ($u$ and $v$) has the smallest measurement error.

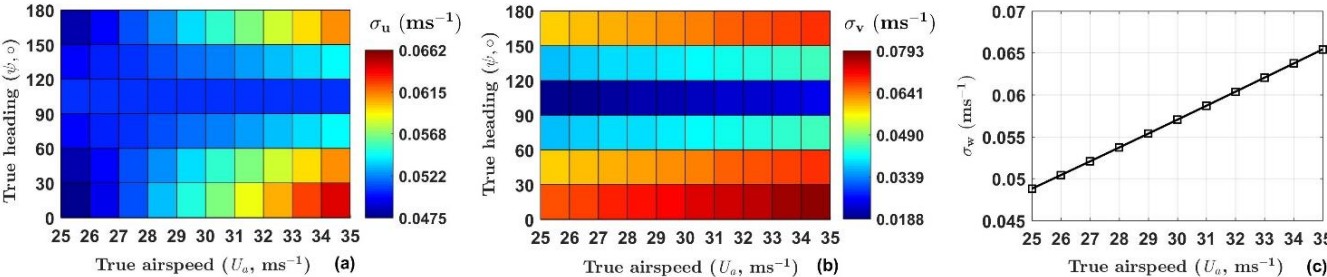


**Figure 4. Estimated measurement precision (1$\sigma$) of the horizontal wind (a, b) and vertical wind (c) according to the equations**
**described in Supplement Part A (Eqs. S18 to S23).**
Generally speaking, an autopiloted UAV can maintain a near-constant true airspeed during the cruise flight phase. For a true
airspeed of 30 m s$^{-1}$ for the current UAV-based EC system during the cruising, the maximum measurement error in the
northward, eastward, and vertical velocities of the geo-referenced wind components were calculated as approximately 0.06,
0.07, and 0.06 m s$^{-1}$, respectively. Then, we assume that a minimum signal-to-noise ratio of 10:1 is required to measure the
wind components with sufficient precision for EC measurements (Metzger et al., 2012). Accordingly, in the real environments,
horizontal and vertical wind speed greater than 0.7 m s$^{-1}$ and 0.6 m s$^{-1}$ can be reliably measured, respectively (Table 2).
**Table 2**: The maximum measurement error in the northward ($u$), eastward ($v$), and vertical ($w$) velocities of the geo-referenced
wind components at the true airspeed of 30 m s$^{-1}$, and the least resolvable magnitude assuming the minimum required signal-
to-noise ratio of 10:1.

| Measurements | Measurement precision (1$\sigma$) | Least resolvable magnitude |
|---|---|---|
| $u$-windspeed (m s$^{-1}$) | 0.06 | 0.6 |
| $v$-windspeed (m s$^{-1}$) | 0.07 | 0.7 |
| $w$-windspeed (m s$^{-1}$) | 0.06 | 0.6 |

The above results gave the nominal precision for wind measurements that does not consider the influence of environmental
conditions. Changes in the environment will lead to sensor drift, increasingly deteriorating the measurement with flight
duration (Metzger et al., 2012; Lenschow and Sun, 2007). Following the methods of Lenschow and Sun (2007), the ability of
the limitations of the accuracy of wind field measurements from UAV to resolve the mesoscale variations of the 3D wind
components in the encountered atmospheric conditions was assessed. For the vertical wind, the mesoscale variability was
defined as the peak signal magnitude of the power spectra curve. The corresponding average wavenumber was determined as
0.09 m$^{-1}$ based on the straight flight leg (about 4 km, lasting about 120 s) of the standard operational flight. Then, the minimum
required signal level for the vertical wind measurement was estimated as $\partial w/\partial t \simeq 0.14$ m s$^{-2}$. The accuracy of the vertical
wind measurement using Eq. (2) is estimated as follows. The first term on the right-hand side of Eq. (2) is dominated by the
drift in the differential pressure transducer, the value of $\partial U_a = 0.05$ m s$^{-1}$ acquired from the wind tunnel test was applied
(Table 1). The histogram of $\Theta$ derived from the standard operational flights is shown in Figure 5. The 99 % confidence interval
indicates that the value of $\Theta$ seldom exceeds $\pm 3°$, i.e., $\pm 0.053$ radians. Thus, the value of the first term was estimated as
$2.2 \times 10^{-5}$ m s$^{-2}$.

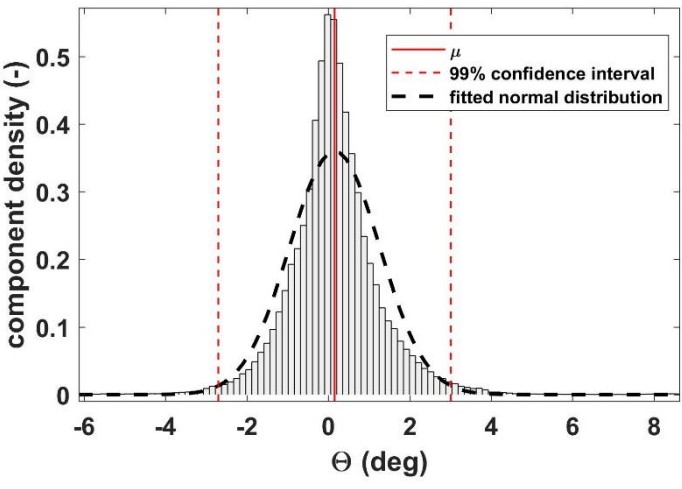


**Figure 5. Histogram of $\Theta$ derived from the standard operational flight. Component density is scaled so that the histogram has a total area of one. Red vertical lines indicate distribution average (solid) and 99% confidence interval (dashed). The black dashed bell curve displays a reference fitted normal distribution.**

The second term in Eq. (2) is a combination of INS pitch accuracy and drift in the measured attack angles. The combined
accuracies of these two sensors were applied to derive $\partial \Theta = 0.0024$ radians. Thus, the second term in Eq. (2) was estimated
as $6 \times 10^{-4}$ m s$^{-2}$. Finally, the third term in Eq. (2) was estimated as $1.7 \times 10^{-4}$ m s$^{-2}$, according to the stated accuracy of
the vertical velocity from the INS. The overall performance of the vertical wind measurement ($7.9 \times 10^{-4}$ m s$^{-2}$) was accurate
enough to resolve the mesoscale variations in vertical air velocity.
The required accuracy of horizontal wind for mesoscale measurement was estimated as 10 times larger than that of vertical
wind, i.e., $\partial u/\partial t \simeq \partial v/\partial t \simeq 1.4$ m s$^{-2}$. The measurement accuracy of the horizontal wind component $u$ was estimated as
$4.8 \times 10^{-4}$ m s$^{-2}$ according to Eq. (3). Like the first term in Eq. (2), with the value of $\Psi$ rarely exceeding $\pm 0.18$ radians, the
measurement accuracy of the horizontal wind component $v$ was estimated as $2.7 \times 10^{-2}$ m s$^{-2}$ according to Eq. (4). Thus, the
measurement accuracy of the horizontal wind components was accurate enough to resolve the mesoscale variations in the
horizontal air velocity as well.
Second, before checking the quality of the acquired calibration parameters, the calibration results of the offset in pitch ($\epsilon_\theta$)
and heading ($\epsilon_\psi$) angles based on the 'box' maneuver are provided in Supplement Part C (Figs. S2 and S3). The final calibration
values are $\epsilon_\theta = -0.183°$ and $\epsilon_\psi = 2°$. In order to verify the quality of these calibration parameters, a 'racetrack' maneuver
was performed. Figure 6 shows the verification results by plotting wind vector and calculating summary statistics for the
'racetrack' maneuver (including turns), using the initial ($\epsilon_\theta = \epsilon_\psi = 0°$, Fig. 6a) and calibrated (Fig. 6b) set of parameters. The
introduction of the calibration parameter effectively improved the quality of geo-referenced wind vector measurement. The
standard deviation for wind direction, $\sigma_{U_{dir}}$, is 4.9° for the calibrated set compared to 8.7° for the initial set, and the standard
deviation of wind speed, $\sigma_U$, is 0.52 m s$^{-1}$ for the calibrated set compared to 1.12 m s$^{-1}$ for the initial set. The average vertical
wind speed is much closer to zero ($\overline{w} = -0.006$ m s$^{-1}$) for the calibrated set than for the initial set ($\overline{w} = 0.1$ m s$^{-1}$). For the
horizontal wind, it is evident from Fig. 6 that the wind direction and velocity are little affected by sharp turns. On the contrary,
the measurement of the vertical wind component is obviously affected by turns in flight, as shown by the large ripple in the
vertical wind speed around the scan value of 150 (Fig. 6). It should be noted that the influence of upwash flow and the leverage
effect are not considered in the calculated of geo-referenced wind vector.

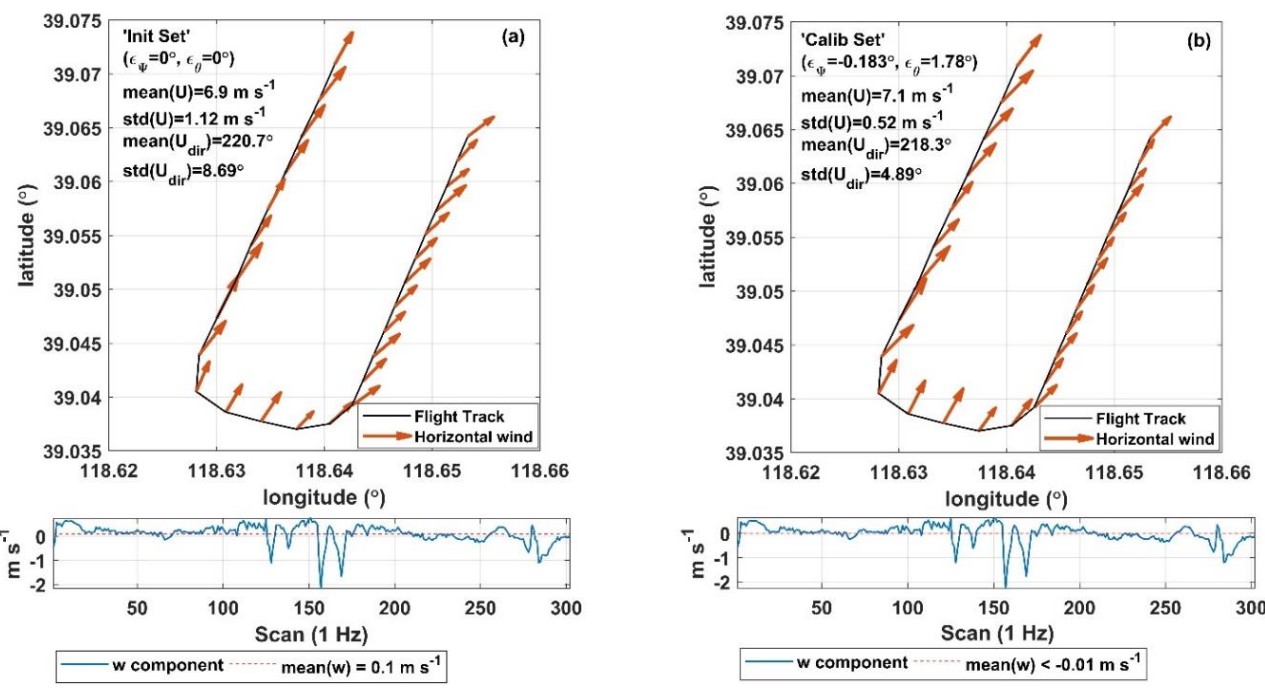


**Figure 6. Quality check of the calibration parameter by plotting wind vector and calculating summary statistics for the 'racetrack'**
**maneuver, using the initial (a) and calibrated (b) set of parameters. The calibration flight was carried out on 4 September 2022 at**
**the Caofeidian Shoal Harbor.**
Third, in order to check the influence of the lift-induced upwash on the attack angle measurement from the 5HP, an
'acceleration-deceleration' flight maneuver was performed. During the 'acceleration-deceleration' maneuver, INS data shown
a vertical velocity of the UAV at 0.05±0.2 m s$^{-1}$, the altitude of UAV at 392±0.6 m, the heading of UAV at 199±2.4°. The
flight conditions met the requirements of the 'acceleration-deceleration' maneuver (Vellinga et al., 2013). The relationship
between the pitch angle ($\theta$) measured by INS and the attack angle ($\alpha$) measured by 5HP is plotted in Figure 7, where the attack
angle was not corrected for lift-induced upwash. The slope (0.94) between $\theta$ and $\alpha$ is close to its theoretical value of 1, and
the intercept (0.16) is close to zero. This result indicates that the lift-induced upwash has only a very small effect on the attack
angle, and the influence of upwash could be ignored.

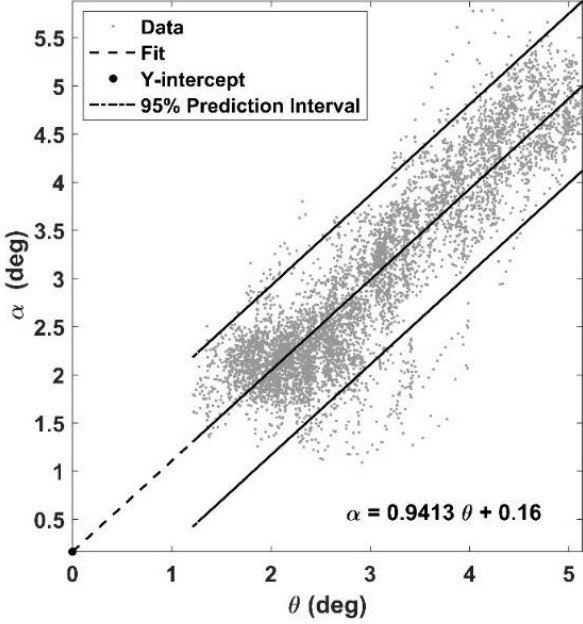


**Figure 7. Relationship between the pitch angle ($\theta$) measured by INS and the attack angle ($\alpha$) measured by 5HP. The fitted linear**
**equation is also shown.**
Finally, the geo-referenced wind vector was calculated with and without the correction for the leverage effect based on the
measurement data from the 'acceleration-deceleration' flight maneuver. The average relative differences between the corrected
and uncorrected horizontal and vertical wind speeds are 0.1 % and 0.2 %, respectively. The standard deviation for horizontal
wind speed is 0.307 m s$^{-1}$ without the level arm term compared to 0.306 m s$^{-1}$ when the level arm term is introduced. The
standard deviation of vertical wind speed is 0.254 m s$^{-1}$ without the level arm term compared to 0.253 m s$^{-1}$ with the level arm
term. The correction of leverage effect had minimal effect on improving the geo-referenced wind vector measurement;
therefore, this correction term can be ignored.
**3.2 Flux measurement error caused by instrumental noise**
Flux measurement error caused by the instrumental noise gives the lowest limit of the flux that the UAV-based EC system is
able to measure. Such measurement error was assessed by combining the covariance uncertainty estimated by RS method and
the propagation of errors in flux correction terms. Before estimating the flux covariance uncertainty using RS method, using
the measured data from each straight and level flight leg of the standard operational flight (Fig. 2), the normalized integrated
cospectra (ogives) curves of sensible heat (Fig. 8a), latent heat (Fig. 8b), and $CO_2$ (Fig. 8c) flux are formed as a function of
wavenumber ($k$), where $k = 2\pi f / U_a$. As shown in Figure 8, although the heterogeneous turbulence (or mesoscale turbulence)
interfered the shape of ogive curves, most curves converged at the high and low frequency ends, which indicated that these
segmented data were sufficiently long to represent the lowest significant frequencies contributing to the covariance.

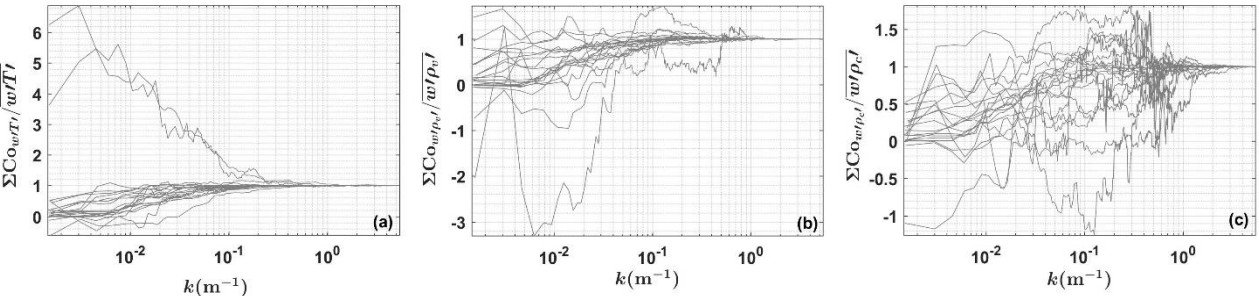


**Figure 8. Normalized ogive curves as a function of wavenumber for the flux covariance of sensible heat (a), latent heat (b), and CO$_2$ (c) from each straight and level flight leg of the standard operational flight in Section 2.2.2.**

Then, the results of instrumental noise related relative flux measurement error compared to the magnitude of the flux are
shown in Figure 9. It can be seen that the flux measurement error caused by instrumental noise significantly decreases when
the flux magnitude increases. It is not surprising since, in theory, instrumental noise is usually close to a constant and the
relative flux measurement error caused by instrumental noise will decreases with increasing measurement magnitude. Overall,
instrumental noise has the least effect on latent heat flux (ranging from 0.02% to 2.42% in this study) measurements, followed
by sensible heat flux (ranging from 0.05% to 8.6% in this study), and has the greatest effect on the measurement of CO$_2$ flux
(ranging from 0.22% to 75.6% in this study). Then, a simple rational function relationship between the relative measurement
error and the flux magnitude is fitted according to the measured data, where the constant term in the denominator is set to 0.
The fitted coefficient in the numerator can be considered as the flux measurement error caused by instrumental noises, which
are 0.03 µmol m$^{-2}$ s$^{-1}$, 0.02 W m$^{-2}$, and 0.08 W m$^{-2}$ for the measurement of CO$_2$ flux, sensible and latent heat flux, respectively.
At last, using the signal-to-noise ratio of 10:1, the minimum magnitudes for reliably resolving the CO$_2$ flux, sensible and latent
heat fluxes were estimated as 0.3 µmol m$^{-2}$ s$^{-1}$, 0.2 W m$^{-2}$, and 0.8 W m$^{-2}$, respectively.

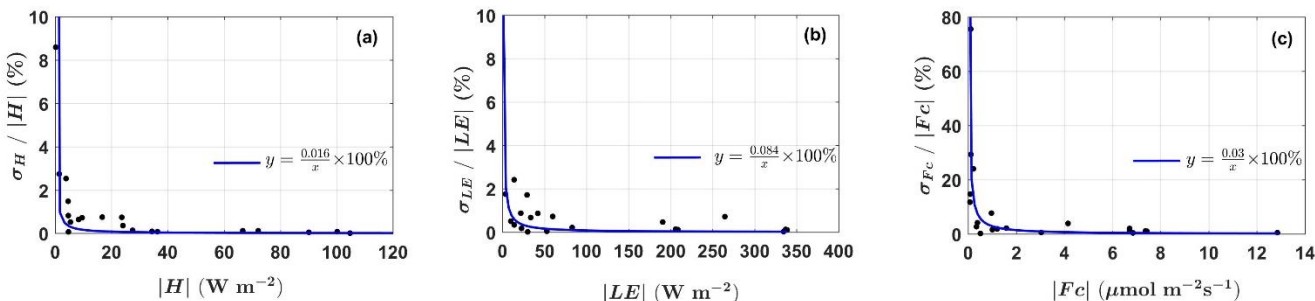


**Figure 9. Relative flux measurement error caused by instrumental noise plotted against the magnitude of the flux. Also shown the fitted error curves. Measured data was from the standard operation flights in Section 2.2.2.**

### 3.3 Resonance noise

The resonance noise from the engine and propeller can lead to systematic overestimation of the variance and covariance of the observed atmospheric scalars. Since the noise mainly appears in the high frequency domain of the (co)spectra, the reference (co)spectral curve of Massman and Clement (2005) was used to quantify the systematically bias caused by the resonance noise.

All spectra curves of the variance of measured scalars (including air temperature, $H_2O$, and $CO_2$ concentration) approximately followed the reference spectra curve and the reference -2/3 slope in the inertial subrange (Figs. 3a to 3c). The largest scatter occurred in the spectra of $CO_2$ (Fig. 3c). When comparing the spectra curve with the reference spectra, the resonance noise led to a systematic deviation in the variance of air temperature, $H_2O$, and $CO_2$ concentration of 0.1±0.1 %, 1.0±0.79 %, and 4.4±0.66 %, respectively, relative to the entire frequency range. For the flux covariance of sensible, latent heat and $CO_2$, all the co-spectra curves approximately follow the reference co-spectra curve and the reference -4/3 slope in the inertial subrange (Figs. 3d to 3f). Compared with the reference co-spectra, the resonance noise led to a systematic deviation in the flux of sensible, latent heat, and $CO_2$ of 0.07±0.004 %,0.3±0.25 %, and 2.9±1.62 %, respectively, relative to the entire frequency range.

The results show that the resonance noise has a very little impact on the measured variance and flux covariance. Among them, the measurements of $CO_2$ concentration and flux are most susceptible to the resonance noise, but the impact of this noise is limited to around 5 % of the observed value.

### 3.4 Sensitivity analysis

In this study, in order to investigate the relevance of the calibration parameters for the measurement of the geo-referenced wind vector and turbulent flux, two sensitivity tests were conducted by adding an error of $\pm 30$ % to the used optimum parameters ($\epsilon_\psi, \epsilon_\theta, \epsilon_\phi, \epsilon_r$). We assumed that the maximum uncertainties contained in the calibration parameter is not more than 30 % of its own value.

First, the sensitivity of the geo-referenced 3D wind and turbulent flux to the uncertainty in the individual parameter was tested. The *RE* value is used to quantify the sensitivity, and the results are summarized in Tables 3 and 4. For the measurement of the geo-referenced wind vector, Table 3 shows that the uncertainties in the temperature recovery factor ($\varepsilon_r$) and 5HP mounting misalignment error in the roll ($\epsilon_\phi$) angle do not contribute significantly to errors in the wind measurements, which were typically smaller than 4% of the observed value in this study. Parameter $\varepsilon_\theta$ had the largest effect on the vertical wind component (up to 30 %), whereas $\varepsilon_\psi$ had the largest effect on the horizontal wind component. For the measurement of turbulent fluxes, Table 4 shows that the errors in $\varepsilon_r$ and $\epsilon_\phi$ does not significantly influence the flux measurements, typically small than 5% of the observed value in this study. The uncertainties in calibration parameter $\varepsilon_\theta$ and $\varepsilon_\psi$ had significant effects on the measurement of turbulent flux. Adding an error of $\pm 30$ % to $\varepsilon_\theta$ result in significant perturbation (large *RE* variance) in the measured turbulent fluxes including sensible heat, latent heat and $CO_2$. While, errors in $\varepsilon_\psi$ to some extent mainly affect the measurement of latent heat flux (*RE* may up to 15 %).

**Table 3**: *RE* of the sensitivity test for the geo-referenced 3D wind vector $(u, v, w)$. An error factor of $\pm30\,\%$ was added to
each calibrated parameter. The geo-referenced 3D wind vector was calculated based on the straight leg of the standard
operational flight.

| Parameter | Error (%) | RE of geo-referenced 3D wind vector mean ± std | | |
|---|---|---|---|---|
| | | $u$ (%) | $v$ (%) | $w$ (%) |
| $\varepsilon_r$ | -30 | 0.04±0.41 | -0.004±2 | 0±0 |
| | 30 | 0.06±0.43 | 0.27±1.1 | -0.07±0.23 |
| $\varepsilon_\varphi$* | -30 | 0.41±2.51 | -0.09±2.05 | 1.15±2.43 |
| | 30 | -0.43±2.61 | 0.09±1.79 | -1.1±2.66 |
| $\varepsilon_\theta$ | -30 | 0.03±0.41 | -0.35±2.54 | -30.51±6.42 |
| | 30 | 0.05±0.45 | 0.42±1.82 | 30.37±6.61 |
| $\varepsilon_\psi$ | -30 | 2.98±25.06 | -2.04±16.3 | 0±0 |
| | 30 | -2.97±24.96 | 2.42±16.63 | 0±0 |

* The optimum calibration value is set to 0, $\varepsilon_\varphi$ was varied over $\pm0.9°$, which is 30 % of its typical range.
**Table 4**: *RE* of the sensitivity test for the turbulent fluxes. An error factor of $\pm30\,\%$ was added to each calibrated parameter.
The turbulent fluxes were calculated based on the straight leg of the standard operational flight.

| Parameter | Error (%) | RE of turbulent flux mean ± std | | | |
|---|---|---|---|---|---|
| | | $Fc$ (%) | $H$ (%) | LE (%) | u* (%) |
| $\varepsilon_r$ | -30 | 1.04±3.04 | -0.76±4.82 | 0.1±0.29 | 0±0 |
| | 30 | -1.0±3.3 | 0.74±4.8 | -0.1±0.29 | 0.2±1.07 |
| $\varepsilon_\varphi$* | -30 | 0.07±1.2 | 0.03±0.7 | 0.15±1.51 | 0.54±1.71 |
| | 30 | -0.14±0.89 | -0.06±0.7 | -0.16±1.46 | 0.12±1.61 |
| $\varepsilon_\theta$ | -30 | -3.27±11.18 | -0.8±9.48 | 0.19±11.91 | -4.08±5.61 |
| | 30 | 2.34±10.52 | -0.44±8.24 | -1.27±9.92 | 3.73±4.53 |
| $\varepsilon_\psi$ | -30 | 1.78±5.18 | -0.73±4.87 | 1.89±13.42 | 0.63±5.75 |
| | 30 | -0.99±3.96 | -0.57±3.26 | 2.66±11.76 | -0.59±4.42 |

* See Table 3.
The second sensitivity test was performed to evaluate the overall interaction between calibration parameters and the
calculation of geo-referenced wind vector and turbulent flux by adding an error of $\pm30\,\%$ to all the calibration members
simultaneously. Tables 5 and 6 provided a summary of the *RE* results from the second sensitivity test. For the measurement of
geo-referenced wind vector (Table 5), adding an error of $\pm30\,\%$ to all the calibration parameters at the same time resulted in
great perturbations to both the horizontal (low *RE* with high variance) and vertical wind components (high *RE* with low
variance). For the measurement of turbulent fluxes, 30% error in the calibration parameters can result in errors in measured
fluxes more than 10%. In addition, Table 6 reveals that the latent heat flux is more sensitivity to the errors in the calibration
parameter than other flux measurement (higher mean and variance of the *RE* compared to other measurements).
**Table 5**: *RE* of the sensitivity test for the geo-referenced 3D wind vector $(u, v, w)$ calculated by adding an error of $\pm 30$ % to
all the calibrated parameter simultaneously. The geo-referenced 3D wind vector was calculated based on the straight leg of the
standard operational flight.

| Parameter | Error (%) | RE of geo-referenced 3D wind vector mean ± std | | |
|---|---|---|---|---|
| | | $u$ (%) | $v$ (%) | $w$ (%) |
| All | -30 | 4.24±27.89 | -3.2±21.1 | -29.35±4.63 |
| | 30 | -4.15±27.46 | 3.55±21.91 | 29.16±4.86 |

**Table 6**: *RE* of the sensitivity test for the turbulent flux calculated by adding an error of $\pm 30$ % to all the calibrated parameter
simultaneously. The turbulent flux was calculated based on the straight flight leg of the standard operational flight.

| Parameter | Error (%) | RE of turbulent flux mean ± std | | | |
|---|---|---|---|---|---|
| | | $Fc$ (%) | $H$ (%) | LE (%) | u* (%) |
| All | -30 | -1.19±10.51 | -0.9±8.06 | 2.71±13.91 | -2.92±8.19 |
| | 30 | -0.49±10.01 | -1.66±5.4 | -6.07±13.24 | 1.74±6.55 |

## 4 Discussions

As one in a new generation of airborne flux measurement platforms, the UAV-based EC system can significantly reduce the
cost of implementing airborne flux measurement campaigns and greatly promote their wide application at regional scales. The
current study aimed to evaluate the basic performance of the UAV-based EC system developed by Sun et al. (2021a) in the
measurement of wind vector and turbulent flux.
First, the wind measurement precision (nominal precision) of the UAV-based EC system was estimated by propagating the
sensor errors to the geo-referenced wind vector using the linearized Taylor series expansions from Enriquez and Friehe (1995) .
The $1\sigma$ precision of geo-referenced wind measurement was estimated to be $\pm 0.07$ m s$^{-1}$, and the least resolvable magnitude
for wind measurement was estimated at 0.7 m s$^{-1}$ by assuming the minimum signal-to-noise ratio of 10:1. The derived wind
measurement minimum resolvable magnitude can be used as a basic reference for wind measurement capability of the UAV-
based EC system, and the measured values of wind vector smaller than the minimum resolvable values should be considered
unreliable. The accuracy of the sensors was also assessed by examining the collected data in the real environment (Lenschow
and Sun, 2007; Thomas et al., 2012). Our results revealed that the overall performance of geo-referenced wind measurement
is sufficient accuracy for resolving the mesoscale variations of the 3D wind components under the encountered atmospheric
conditions. Therefore, it is possible to capture the mesoscale variability of the atmospheric boundary layer (ABL) over a wide
range of spatial scales by performing longer flight paths.
Second, based on the measurement data from the in-flight calibration campaign, several key factors affecting the accuracy
of geo-referenced wind measurement were analysed. First, the UAV-based EC system was calibrated (in Supplement Part C)
using measured data from the 'box' flight maneuver to correct the mounting misalignment between the 5HP and the CG of the
UAV in the heading ($\epsilon_\theta = -0.183°$) and pitch ($\epsilon_\psi = 2°$) angles. The quality of the acquired calibration parameters was
verified using the 'racetrack' flight maneuver, and the acquired calibration value effectively improved the observed wind field
with smaller variance compared with the wind calculated using their initial value. At the same time, the measurement of the
vertical wind component was significantly affected by the in-flight turn (maintaining about 20° roll). Therefore, it is necessary
to avoid using the measured data from the turn section for turbulent flux calculation. Compared to other studies (Vellinga et
al., 2013; Reineman et al., 2013), the relatively large variance in the horizontal wind and wind direction after calibrated in this
study may be caused by the nonstationary condition of the turbulence. This was caused by the reason that the flight altitude of
400 m was not high enough to totally avoid interaction from the underlying surface.

618       The current calibration procedure did not include methods to determine the offset angle in roll ($\varepsilon_\varphi$) and the temperature

recovery factor ($\varepsilon_r$) because of the small vertical separation (27.3 cm) between the 5HP and the roll axis of the UAV and the
small Mach number (<0.1) during operational flight. The default ($\varepsilon_\varphi = 0°$) and empirical ($\varepsilon_r = 0.82$) value were adopted for
these two calibration parameters. The sensitivity analysis shown these two parameters have no large effect on the wind vector
and turbulent flux.

623       It should be noted that wind measurements from the airborne platform may be susceptible to flow distortion and rigid-body

rotation (leverage effects). Generally, the influence of these two factors were ignored by UAV platform when calculating the
geo-referenced wind vector. To confirm that these effects could be safely ignored, data from 'acceleration-deceleration' flight
maneuver was used to analyse the effects of lift-induced upwash and the leverage effect on the wind measurements. Our results
demonstrate that the upwash has almost no effect on the wind measurement, which was indicated by the near 1:1 relationship
(0.94 in Fig. 7) between the measured attack angles and pitch angle. The slight departures from the ideal 1:1 relationship may
have been caused by the nonstationary condition during the flight. For the influence from the leverage effects, the differences
in 3D wind vector between corrected and uncorrected for the leverage effect is very small. Thus, ignoring the influence of the
leverage effect has almost no effect on the measurement of wind. Therefore, we concluded that the geo-referenced 3D wind
vector can be measured reliably by the current UAV-based EC system without considering the interference from the lift-
induced upwash and leverage effects.

634       Third, instrumental noise related relative flux measurement error was estimated by combining the covariance uncertainty

estimated by RS method and the propagation of errors in flux correction terms. By assuming that the instrumental noise is
close to a constant, we fitted a simple rational function relationship between the relative measurement error and the flux
magnitude according to measured data (Fig. 9), and the fitted coefficient in the numerator can be considered as the flux
measurement error caused by instrumental noises. The estimated flux measurement error of $CO_2$, sensible and latent heat flux
are 0.03 µmol m$^{-2}$ s$^{-1}$, 0.02 W m$^{-2}$, and 0.08 W m$^{-2}$, respectively. Since the RS method directly uses the shuffled raw
measurement data to calculate the instrumental noise induced flux measurement error, its estimated results inevitably included
the effects of resonance noise from the UAV. Using the signal-to-noise ratio of 10:1, the least resolvable magnitude for
turbulent flux measurement was estimated to be 0.3 µmol m$^{-2}$ s$^{-1}$ for the $CO_2$ flux, 0.2 W m$^{-2}$ for the sensible heat flux, 0.8 W
m$^{-2}$ for the latent heat flux, respectively.
Fourth, because the UAV-based EC system has not completely insulated the noise from the operation of the engine and
propeller and its effect on the measured scalars, the reference (co)spectra of Massman and Clement (2005) was used to quantify
the effect of the resonance noise on the variance and flux of the measured scalars. Previous studies found that the influence of
resonance noise mainly appears in the high frequency domain of the power spectra of the measured atmospheric scalars (e.g.,
air temperature, $H_2O$, and $CO_2$ concentration). The frequency range of the noise region was artificially designated for air
temperature, water vapor and $CO_2$. By calculating the area difference of the designated frequency range beneath the
(co)spectral curve between the measured and reference (co)spectral curves, the resonance effect could be quantified. The
results shown that, overall, resonance noise has little impact on the variance and flux covariance of the measured scalars. The
measurements of $CO_2$ concentration and its flux covariance were the most susceptible to resonance noise, but the maximum
effect was less than 5 %. It should be noted that this method may overestimate the deviation caused by resonance noise as
indicated by the reference (co)spectra curve and the measured (co)spectra not fully overlapping in the inertial subrange (shown
in Fig. 3).
In general, gas detection based on optical absorption methods can achieve fast and high precision gas concentration
measurements, but they are extremely sensitive to vibration noise. However, due to the limited space inside the UAV, it is
difficult to install all the hardware needed for a complex vibration isolation structure to effectively isolate the impact of
vibration on the gas analyser. The weight and the aerodynamic shape of the UAV also present challenges. In the future, a new
UAV-based EC system based on a pure electric UAV will be developed. The electro-powered UAV has similar performance
to the current fuel-powered UAV but can minimize the impact of vibration noise from the engine and propeller rotation, which
makes it possible to completely isolate the resonance effect using a simple vibration isolation structure. Electro-powered UAVs
also have other advantages including larger wingspan (lower cruising speed), a constant CG position, and lower operational
complexity compared to the current system.
Fifth, two sensitivity tests were conducted to assess the perturbation of the geo-referenced wind velocity and turbulent flux
under variation ($\pm 30$ %) of each calibration parameter around its optimum value ($\epsilon_\psi = 2°, \epsilon_\theta = -0.183°, \epsilon_\phi = 0°, \epsilon_r = 0.82$)
as well as under simultaneous variation ($\pm 30$ %) of all calibration parameters. Their RE was used to evaluate the sensitivity,
and values of wind and flux less than their least resolvable magnitude were removed from the calculation. The results revealed
that uncertainties in the temperature recovery factor ($\varepsilon_r$) and mounting offset in roll angle ($\varepsilon_\varphi$) do not significantly contribute
to an error in the measurement of wind vector and turbulent fluxes. The typical RE for the geo-referenced wind measurements
is less than 1.2 % with variance less than 3 %, and the typical RE for turbulent flux is less than 1.1 % with variance less than
5 %. Calibration parameters that had the largest effect on the measurement of geo-referenced wind vector and turbulent flux
are the mounting offset angle in pitch ($\varepsilon_\theta$) and heading ($\varepsilon_\psi$). Uncertainties in $\varepsilon_\theta$ had a direct effect on the measurement of
vertical wind component, and then these errors propagate to the measured fluxes, resulting in a large error contains in the
measured fluxes (∼15 %). A negative error in $\varepsilon_\theta$ will lead to an underestimation of the vertical wind and vice versa. Errors in
$\varepsilon_\psi$ directly affect the measurement of the horizontal wind, and to some extent, the measurement of turbulent flux. The
difference is that the added error in $\varepsilon_\psi$ lead to a great variability (up to 25 %) in the *RE* of horizontal wind. By checking the
relationship between the magnitude of the horizontal wind $(u, v)$ and *RE*, a near exponential relationships was seen, as shown
in Figure 10. The influence of the error in the $\varepsilon_\psi$ decreased significantly with the increase in the magnitude of the horizontal
wind velocity. Additionally, the measurement of latent heat flux may be greatly affected by the error in $\varepsilon_\psi$, which is reflected
by the relatively large deviancy (∼14 %) of the *RE*. Therefore, calibration parameter $\varepsilon_\theta$ and $\varepsilon_\psi$ need to be carefully calibrated.

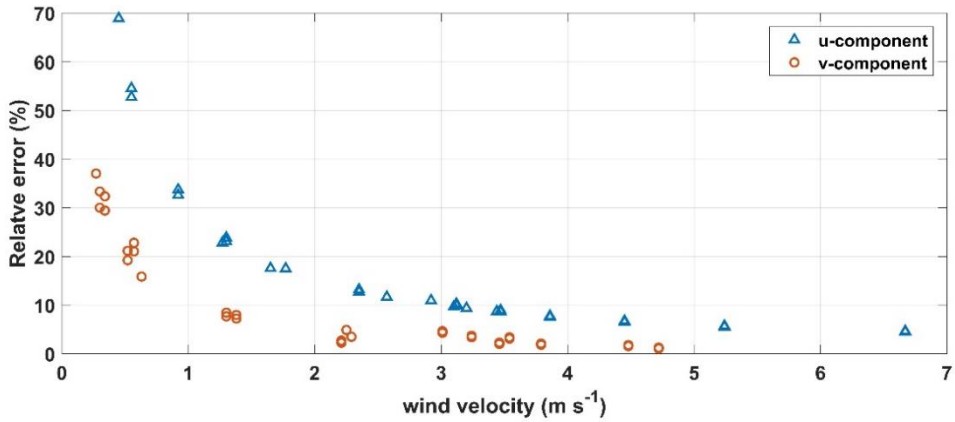


**Figure 10. Relationship between the magnitude of the horizontal wind velocity $(u, v)$ and *RE* from the sensitivity test.**

Lastly, it should be noted that the accuracy of the measured geo-referenced wind vector and turbulent flux from the UAV-
based EC system is subject to the combination of many factors, mainly including sensor accuracy, UAV powerplant, UAV
fluctuation (e.g., variation of the UAV attitude and flight height), and the atmospheric conditions during the measurements,
etc. This study mainly focused on assessing the effects of sensor precision and UAV powerplant on the measurement errors of
geo-referenced wind vector and turbulent flux. Evaluation results gave the lowest limit of the wind vector and turbulent flux
that the UAV-based EC system can measure. Another effective way to evaluate the measurement accuracy of this new
technique is by comparing measured values with those from the traditionally recognized measurement. However, the direct
comparison of flux measurements between aircraft and traditional ground tower is still challenging due to the difference in the
measurement height, mechanism (time series for ground EC and space series for aircraft), and instruments (e.g., wind sensor).
Previous studies have extensively compared the measurement of fluxes and wind vector between airborne and ground-based
EC methods and found consistent results (Gioli et al., 2004; Metzger et al., 2012; Sun et al., 2021b). At the same time,
substantial and consistent over- or underestimation of the measured wind and fluxes by UAV compared to ground
measurements were observed and reported. These differences may be due to several factors such as vertical flux divergence
(the measurement height of UAV is higher than ground-tower), surface heterogeneity (induced by the larger footprint region
of the UAV compared to the ground tower), measurement errors (e.g., window length, resonance noise, etc.) as well as their
difference in platform and sensors. Therefore, it is necessary to conduct a comparison test on the same platform and under the
same environment to exclude the influence of these factors. Inspired by Reineman et al. (2013), future work can include
developing a ground-vehicle-based UAV flux validation platform. This platform could carry both the UAV-based and
traditional ground EC system to assess the measurement accuracy of the UAV-based EC system with the measurement of
ground EC as the benchmark in a flight-like scenario.

## 5 Conclusions and further works

The main objective of this study was to quantitatively evaluate the performance of the developed UAV-based EC system in
the measurement of geo-referenced wind vector and turbulent flux. In terms of measuring precision, turbulence measurements
from the UAV-based EC system were achieved with sufficient precision to enable reliable measurement of geo-referenced
wind and EC flux. Magnitudes larger than 0.7 m s$^{-1}$ for wind velocity, 0.3 µmol m$^{-2}$ s$^{-1}$ for $CO_2$ flux, 0.2 W m$^{-2}$ for sensible
heat flux, and 0.8 W m$^{-2}$ for latent heat flux could be reliably measured by the UAV-based EC system by assuming the
minimum required signal-to-noise ratio of 10:1 for EC application. Based on the data from the calibration flight, the carefully
calibrated offset angle in pitch ($\epsilon_\theta$) and heading ($\epsilon_\psi$) were shown to effectively improve the quality of wind field measurements,
and the influences of flow distortion and the leverage effect on the wind measurement were minimal and could be ignored.
The influence of resonance noise was small on the measurement of air temperature and water vapor (typically < 1 % for their
variance and flux covariance), but relatively large on the measurement of $CO_2$ (around 5 % for variance and flux covariance).
The relevance of the calibration parameters ($\varepsilon_r, \epsilon_\phi, \varepsilon_\psi, \varepsilon_\theta$) for the measurement of the geo-referenced wind vector and
turbulent flux was also assessed based on two sensitivity tests. The measurements of the geo-referenced wind vector and
turbulent flux were insensitive to the errors in the $\varepsilon_r$ and $\epsilon_\phi$. Uncertainties in the calibration parameter $\varepsilon_\theta$ and $\varepsilon_\psi$ had the
strongest effects on the measurements. Because of $\varepsilon_\theta$ directly determining the magnitude of the vertical wind, its error will
lead to uncertainties in vertical wind measurement and then propagate the uncertainties to the measurement of turbulent flux.
The uncertainties in $\varepsilon_\psi$ have a direct effect on the measurement of horizontal wind, and to some extent, the measurement of
turbulent flux. Therefore, these two calibration parameters need to be carefully calibrated. Conducting the UAV-based EC
measurement when wind velocity is larger than 2 m s$^{-1}$ can led to more stable and reliable (*RE* < 20%) results of the wind
speed measurement compared to a relatively windless environmental.
Finally, we concluded that the developed UAV-based EC system measured the geo-referenced wind vector and turbulent
flux with sufficient precision. The lift-induced upwash and leverage effect had almost no effect on the measurement of geo-
referenced wind vector. The resonance effect caused by the operation of engine and propeller mainly affected the measurement
of $CO_2$, and its effect on variance and flux covariance was around 5 %. The quality of calibration parameters $\varepsilon_\psi$ and $\varepsilon_\theta$ has a
significant effect on the measurement of the geo-referenced wind vector and turbulent flux, underscoring the importance of
careful calibration. Although UAV-based EC measurements have many advantages over manned aircraft and tower-based EC
measurements, airborne EC measurements themselves have some shortcomings, such as flux results hard to interpret (e.g.,
influence from surface heterogeneity, flux divergence, etc.), the measurements are restricted to short periods of time, and the
interaction between the UAV and turbulence. Future researches may include the development of a new generation UAV-based
EC system with the following improvements: 1) a new electro-powered UAV platform with the advantages of being quieter
(low noise), having a low cruising speed, and being easy to operate; 2) a ground-vehicle-based validation platform to enable
direct comparative evaluation of the UAV-based EC system with traditional ground EC methods under near-identical
environmental conditions; 3) a graphics based real-time monitoring system to make it possible to change the flight pattern
according to real-time data; and 4) a number of integrated field observation experiments that combining ground-based EC
networks, OMS, and multi-source satellite RS to further prompt the development of theory and methodology for scaling
transformation. Ultimately, the versatility of the UAV-based EC system as a low cost and widely applicable environmental
research aircraft facilitates further improving our understanding of the energy and matter cycling processes at regional scales.
***Author contributions.*** SY, GB and LX planned the field campaign; SY, LB, JJ, ZZ and JS carried out the field measurements.
SY, LS and XZ analysed the data and wrote the manuscript draft. SB, and QZ reviewed and edited the manuscript.
***Competing interests.*** The authors declare that they have no conflict of interest.
***Acknowledgments.*** This work was supported by the National Natural Science Foundation of China (Grant No. 42101477). We
would like to thank F-EYE UAV Technology Co. Ltd. for building, maintaining, and operating the UAV in this study.
***Data availability.*** Data for this research are not publicly available due to its proprietary nature currently. The UAV calibration
flight data and the standard operation flight data in this study are available upon request to the corresponding author.

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
