# Peer review of "Evaluation of the quality of a UAV-based eddy covariance system for"

_Atmospheric Measurement Techniques, 2022_

## Author Comment (AC1)

**Referee #1**

We are truly grateful to your critical comments and thoughtful suggestions. In accordance with the comments, the manuscript has been thoroughly revised in content; the revisions have been marked in red. All references to figure(s), table(s), section(s), page(s), and line(s) refer to the revised manuscript unless otherwise stated.

**General Comments:**

*The authors are to be commended for confronting a major challenge. I believe in much of their introduction regarding the need to increase the spatial density of flux observations, and the potential for UAV platforms to fulfill this need. The paper contains much methodological detail regarding UAV wind measurements and assessment, aspects of the paper that I do not feel qualified to evaluate. However, I can assess eddy covariance methodologies for determining surface fluxes, and I am afraid that the authors as yet fall short regarding both the implementation and error assessment.*

*From reading the text, it is unclear to me whether the authors have (1) incorrectly determined the fluxes of $CO_2$ and $H_2O$, or simply (2) incorrectly described the methodology that they applied. I suspect the former, based on the comment below regarding the text at line 479. In any event, the paper requires major revision to clarify these points, and possibly to modify both the eddy covariance methodology and also the assessment of its errors and sensitivity to environmental parameters. I also believe that the presentation of the data could be improved significantly as described below.*

Re: Thank you for your insightful comments. Most of the above comments are handled more specifically bellow. We have substantially revised this manuscript in both the methodology for error assessment and the relative contents. In particular, aspects involving the calculation of turbulent fluxes (including the necessary corrections) and the error analysis of wind and flux measurements have been thoroughly revised. Your comments are very helpful to improve the quality of the manuscript.

**Specific Comments**

*Q1. 82 - As the authors note "The EC method is a well-developed technology for directly measuring vertical turbulent flux...". Decades of experience that has shown that the covariances between the vertical wind and the densities of $CO_2$ ($\rho c$) and $H_2O$ ($\rho v$) - measured directly by the EC150 - do not define the turbulent fluxes of these gases, as the authors seem to believe (lines 250-251). This is because fluctuations in these variables are predominantly caused by temperature fluctuations (due to heat exchange), and $\rho c$ also fluctuates because of varying humidity (due to evaporation). See comment regarding line 269 below.*

Re: The original sentence (Lines 251-253 in the original manuscript) for describing the method of EC flux measurement may be not appropriate, and it created ambiguity for readers. In this study, the calculation of turbulent fluxes, especially for latent heat and $CO_2$ flux, included necessary correction for air density fluctuations (WPL correction). In order to clearly express the methodology for calculation the turbulent fluxes by UAV-based EC system in this study, we modified and added necessary descriptions in the revised manuscript and supplement materials as follows:

Lines 251-253, in the revised manuscript, the original sentence is revised to "In the final stage, based on the EC technology and spatial averaging, the turbulent flux is calculated using the covariances of vertical wind ($w$) with air temperature ($T_a$) for sensible heat flux ($H$), with water vapor density ($q$) for latent heat flux (LE), and with $CO_2$ density ($c$) for $CO_2$ flux ($F_c$), and with the necessary correction."

Lines 256-256, in the revised manuscript, we added the follow text: "Detailed calculation procedure and formulas of $H$, LE, and $F_c$ used by the present UAV-based EC system are provided in Supplement Part B, including spatially averaging, coordinate rotation, and necessary correction (i.e., WPL correction for LE and $F_c$)."

In the revised Supplement Part B, this section provided a detailed description of the process and methodology for calculating turbulent fluxes and error analysis.

*Q2. 162 - The choice of the Bohai Sea as the place for the in-flight calibration campaign is quite unfortunate. Its waters are cool, particularly relative to continental temperatures in September, and therefore the magnitude and spectra of the temperature fluctuations that tend to dominate fluctuations in ρc and ρv are not representative of what might be encountered in many other environments. Indeed, the authors note that stable atmospheric conditions prevailed during the campaign (lines 174-175), implying that turbulence is supressed during this assessment of the ability to measure turbulent fluxes. If this limitation cannot be removed from the analysis, it should at least be noted.*

Re: The main objective of calibration flight is to acquire the mounting misalignment angle in the heading ($\epsilon_\psi$) and pitch ($\epsilon_\theta$) between the 5HP (five-hole probe) and the CG (center of gravity) of the UAV. The calibration flight should be carried out under specific atmospheric conditions to ensure a continuous, stable and ground-independent wind component.

The common assumptions for calibration flight include 1) low turbulence or turbulent transport (i.e., no disturbance), 2) a constant mean horizontal wind, and 3) mean vertical wind near zero (Drüe and Heinemann, 2013; Vellinga et al., 2013; Van Den Kroonenberg et al., 2008) (Lines 177-180). These assumptions are usually well satisfied above the ABL or under stable atmospheric conditions. Over the sea surface, due to its uniform and cool surface property, the turbulence fluctuations are weaker than that over the land surface, making where a more ideal environment to conduct calibration flight (Lines 180-183). Accordingly, we revised the original sentence (Lines 166-169, in the original manuscript) as follows:

Lines 177-183, in the revised manuscript: "The assumptions for calibration flight include 1) low turbulence or turbulent transport (i.e., no disturbance), 2) a constant mean horizontal wind, and 3) mean vertical wind near zero (Drüe and Heinemann, 2013; Vellinga et al., 2013; Van Den Kroonenberg et al., 2008). This allows identical wind components for several consecutive straights in opposite or vertical flight directions. These assumptions are usually well satisfied above the ABL or under stable atmospheric conditions (Drüe and Heinemann, 2013). Over the sea surface, due to its uniform and cool surface property, the turbulence fluctuations are weaker than that over the land surface (Mathez and Smerdon, 2018), making where a more ideal environment to conduct calibration flight."

*Q3. 261-262 - "In this study, the objective is not to quantify the actual flux exchange between the surface and the atmosphere, but rather to assess the sensitivity of the calculated turbulent flux to external parameters." The quality of a UAV-based eddy covariance system for measurements turbulent flux (reflecting the title of the paper) cannot be assessed without determining whether it quantifies the actual surface exchange. As an example, if the system reports an unbelievable uptake of 50 μmol m-2 s-1 of CO2 uptake, its quality is likely low whatever its sensitivity to external parameters. For this reason, I believe that the authors should indeed provide magnitudes of the fluxes that they are characterizing. This is particularly so given methodological uncertainties regarding how the fluxes are determined (see comment regarding line 82 above, and 269 below).*

Re: In accordance with your comment, when analyzing the flux measurement error, we also provided the magnitudes of the fluxes. In Section 3.2 of the revised manuscript, the relationship between the estimated relative flux measurement error and the flux magnitude was show in Figure 9. Accordingly, the original sentence (Lines 261-262 in the original manuscript) was removed.

*Q4. 269 - The authors cite Metzger et al. (2012) regarding the calculation of turbulent fluxes. Since Metzger et al. (2012) did not measure CO2 fluxes, the citation at line 269 is insufficient to document the data processing and corrections needed to determine the turbulent CO2 flux. Users of an open-path IRGA must address the issue of "density corrections", and cite an appropriate reference (Webb et al., 1980, Correction of flux measurements for density effects due to heat and water vapor transfer. Quart J Roy Meteorol Soc 106:85–100; or perhaps Kowalski et al.,2021, Disentangling Turbulent Gas Diffusion from Non-diffusive Transport in the Boundary Layer. Boundary-Layer Meteorol 179, 347–367. https://doi.org/10.1007/s10546-021-00605-5). Otherwise, the results are inconsistent with what we know about biological activity.*

Re: In the revised manuscript (Lines 256-258) and Supplement Part B, we supplemented the detailed calculation procedure and formulas of sensible heat, latent, and $CO_2$ flux, including spatially averaging, coordinate rotation, and necessary correction (i.e., WPL correction for LE and $F_c$). Inaccuracies in literature citations have also been corrected (Lines 133-135).

*Q5. 273-278 - There is a problem with randomly generated errors for input variables, because some of these variables tend to be correlated. For example, over the sea in stable atmospheric conditions, the heat flux is downward and the vapor flux is presumably upward. Therefore,*

*temperature and humidity are negatively correlated. If, for some reason, errors in the measurement of temperature and humidity are correlated, then this can badly bias the eddy covariance. Correlated errors could arise for many reasons including faulty instrumentation, sampling errors, and flow distortion.  A Monte Carlo simulation that presumes independence of such variables will miss this sort of problem, and therefore is not an appropriate tool for error assessment.*

Re: In the revised manuscript, the original Monte Carlo error simulation method used to estimate the measurement error of geo-referenced wind and turbulent flux has been removed. Instead, we used the partial derivatives of the full calculation equation for geo-referenced wind and turbulent flux to estimate the measurement error in wind and fluxes. The detailed methods and procedures to estimate the measurement error of geo-referenced wind vector and turbulent fluxes were gave in Supplement Part A and Part B.

For estimating the measurement precision of geo-referenced wind, in the revised manuscript, we used the linearized Taylor series expansions derived by Enriquez and Friehe (1995) (in the revised Supplement Part A) to determine the sensitivities of each of the geo-referenced wind vector components with respect to the relevant variables. Then, combined these sensitivity terms to estimate the overall measurement error ($1\sigma$) in the geo-referenced 3D wind vector. The results were provided in Section 3.1 of the revised manuscript. It concluded that the measurement precision for geo-referenced wind vector is related to the true airspeed and heading of the UAV (Lines 425-437). For a true airspeed of 30 m s$^{-1}$ for the current UAV-based EC system during the cruising, the maximum measurement error in the northward, eastward, and vertical velocities of the geo-referenced wind components were calculated as approximately 0.06, 0.07, and 0.06 m s$^{-1}$, respectively (Lines 438-441).

For flux measurements, in this study, we mainly focused on the error caused by instrumental noise due to they are related not only to the system performance, but also to the minimum resolvable capability for the flux to be measured. In the revised manuscript, we added a section (Section 2.4.2) to illustrate the methods for estimating flux measurement error caused by instrumental noise by combining the covariance uncertainty estimated by RS method (Eq. 6 in the revised manuscript) and the propagation of errors in flux correction terms (Eqs. S29-S31 in Supplement Part B). In the revised Supplement material Part B, we gave the detailed equations to calculate the fluxes of sensible heat, latent heat, carbon dioxide ($CO_2$), and the method to quantify the measurement uncertainty in them due to instrument noise. The results were given in Section 3.2 of the revised manuscript, and the flux measurement error caused by instrumental noises was estimated at 0.03 µmol m$^{-2}$ s, 0.02 W m$^{-2}$, and 0.08 W m$^{-2}$ for the measurement of $CO_2$ flux, sensible and latent heat flux, respectively (Lines 522-536).

*Q6. 396 - As noted above (see comment regarding line 273), the method used to determine the least resolvable flux magnitude is not believable.*

Re: Please see the answers to Q5. In the revised manuscript, we used the partial derivatives of the full calculation equation for geo-referenced wind and turbulent flux to estimate the measurement

error in wind and fluxes. These partial derivative equations were given in Supplement Part A and Part B. Accordingly, we assumed a minimum required signal-to-noise ratio of 10:1, and estimated the least resolvable wind speed and flux magnitude. Accordingly, the text was modified as follow:

Lines 441-443, in Section 3.1 of the revised manuscript, we gave the results of the estimated least resolvable wind speed: "Then, we assume that a minimum signal-to-noise ratio of 10:1 is required to measure the wind components with sufficient precision for EC measurements (Metzger et al., 2012). Accordingly, in the real environments, horizontal and vertical wind speed greater than 0.7 m $s^{-1}$ and 0.6 m $s^{-1}$ can be reliably measured, respectively (Table 2)."

Lines 532-533, in Section of the revised manuscript, we gave the results of the estimated least resolvable flux magnitude: "At last, using the signal-to-noise ratio of 10:1, the minimum magnitudes for reliably resolving the $CO_2$ flux, sensible and latent heat fluxes were estimated as 0.3 µmol $m^{-2}$ s, 0.2 W $m^{-2}$, and 0.8 W $m^{-2}$, respectively."

***Q7. 399 - Regarding sensor drift, the authors should take care to examine the effects of any lens contamination (Serrano-Ortiz et al., 2008, Consequences of uncertainties in CO2 density for estimating net ecosystem exchange by open-path eddy covariance, Boundary-Layer Meteorology, 126, 209-218.), which would seem to be a problem in an environment rich in sea salt. To be clear, such errors arise as a consequence of the necessary density corrections when measuring gas densities with an open-path IRGA.***

Re: Thanks a lot for this comment. UAV EC flux measurements do not need to take into account of the problem of lens contamination due to the signal quality of the IRGA was checked before each flight measurement to ensure that the measurement of gas concentration is not affected by lens contamination. Accordingly, we added the necessary explanation text in the revised manuscript as follow:

Lines 361-362, in Section 2.4.2: "For EC measurement from our UAV, the signal quality of the IRGA is checked before each flight measurement to ensure that the measurement of gas concentration is not affected by lens contamination."

***Q8. 479 - "a sensitivity test was conducted by adding an error of ±30 % to the calibrated value of each calibration parameter." Serrano-Ortiz et al. (2008) showed that just a 5% in the CO2 density can cause CO2 flux errors in excess of 13%, due to the influence of density corrections. The fact that the authors of this study found such small errors in the CO2 flux (Table 4) strongly hints that they are not correcting for density effects, and therefore that their error analysis is inadequate. It also causes me to strongly doubt the claim of a 0.4 µmol m-2 s-1 least resolvable magnitude for the CO2 flux.***

Re: The main objective of the sensitivity test is to understand the relevance of the calibration parameters for the measurement of geo-referenced wind vector and turbulent flux. Four calibration parameters included in the sensitivity test, including three mounting misalignment angles ($\epsilon_\psi, \epsilon_\theta, \epsilon_\phi$) between the 5HP and the CG of the UAV and one temperature recover factor ($\epsilon_r = 0.82$). The

reliability of these calibration parameters directly affects the uncertainty of wind measurement and then indirectly affects the uncertainty of flux measurements (Vellinga et al., 2013). The sensitivity test method used in this study was similar to Vellinga et al. (2013), but in order to highlight the perturbation affected by the uncertainty in calibration parameters, an error of $\pm 30\%$ was added to their optimum value (Section 2.4.4 in the revised manuscript).

Serrano-Ortiz et al. (2008) analysed the error relevance between the measurement of $CO_2$ density and $CO_2$ flux, however, this study analysed the error relevance between the acquired calibration parameters $(\epsilon_\psi, \epsilon_\theta, \epsilon_\phi, \epsilon_r)$ and wind measurement as well as flux. As mentioned above, these calibration parameters do not directly affect the precision of flux measurements. The claim of a 0.4 $\mu mol\ m^{-2}\ s^{-1}$ least resolvable magnitude for the $CO_2$ flux (Line 397 in the original manuscript) was revised to 0.3 $\mu mol\ m^{-2}\ s^{-1}$ according to new error analysis method (Section 3.2 in the revised manuscript and Supplement Part B) in the revised manuscript. In the revised manuscript, these small errors or claimed least resolvable magnitudes for flux measurement are related only to instrument noise. Generally speaking, the effect of instrumental noise on the uncertainty of flux measurement is very small (Metzger et al., 2012; Mauder et al., 2013).

*Q9. 583 - Change "Forth" to "Fourth".*

Re: This mistake has been revised in the revised manuscript.

**References:**

Drüe, C. and Heinemann, G.: A Review and Practical Guide to In-Flight Calibration for Aircraft Turbulence Sensors, Journal of Atmospheric and Oceanic Technology, 30, 2820-2837, 10.1175/JTECH-D-12-00103.1, 2013.

Enriquez, A. G. and Friehe, C. A.: Effects of Wind Stress and Wind Stress Curl Variability on Coastal Upwelling, Journal of Physical Oceanography, 25, 1651-1671, https://doi.org/10.1175/1520-0485(1995)025<1651:EOWSAW>2.0.CO;2, 1995.

Mathez, E. and Smerdon, J.: Climate Change3. Ocean– Atmosphere Interactions, in: The Science of Global Warming and Our Energy Future, Columbia University Press, 69-100, doi:10.7312/math17282-005, 2018.

Mauder, M., Cuntz, M., Drüe, C., Graf, A., Rebmann, C., Schmid, H. P., Schmidt, M., and Steinbrecher, R.: A strategy for quality and uncertainty assessment of long-term eddy-covariance measurements, Agricultural and Forest Meteorology, 169, 122-135, https://doi.org/10.1016/j.agrformet.2012.09.006, 2013.

Metzger, S., Junkermann, W., Mauder, M., Beyrich, F., Butterbach-Bahl, K., Schmid, H. P., and Foken, T.: Eddy-covariance flux measurements with a weight-shift microlight aircraft, Atmos. Meas. Tech., 5, 1699-1717, 10.5194/amt-5-1699-2012, 2012.

Serrano-Ortiz, P., Kowalski, A. S., Domingo, F., Ruiz, B., and Alados-Arboledas, L.: Consequences of Uncertainties in $CO_2$ Density for Estimating Net Ecosystem $CO_2$ Exchange by Open-path Eddy Covariance, Boundary-Layer Meteorology, 126, 209-218, 10.1007/s10546-007-9234-1, 2008.

van den Kroonenberg, A., Martin, T., Buschmann, M., Bange, J., and Vörsmann, P.: Measuring the Wind Vector Using the Autonomous Mini Aerial Vehicle M2AV, Journal of Atmospheric and Oceanic Technology, 25, 1969-1982, 10.1175/2008JTECHA1114.1, 2008.

Vellinga, O. S., Dobosy, R. J., Dumas, E. J., Gioli, B., Elbers, J. A., and Hutjes, R. W. A.: Calibration and Quality Assurance of Flux Observations from a Small Research Aircraft*, Journal of Atmospheric and Oceanic Technology, 30, 161-181, 10.1175/JTECH-D-11-00138.1, 2013.

---

## Author Comment (AC2)

**Referee #2**

We are truly grateful to your critical comments and thoughtful suggestions. In accordance with the comments, the manuscript has been thoroughly revised in content; the revisions have been marked in red. All references to figure(s), table(s), section(s), page(s), and line(s) refer to the revised manuscript unless otherwise stated.

**General Comment**

*This paper presents a systematic assessment results of a UAV-based eddy covariance (EC) system developed by Sun et al. (2021) on the measurement ability in wind and turbulent flux. Overall, the objectives are clearly put forward and well-motivated. The UAV EC system itself is novel and interesting, and the topics are closely related to the current research hotspots.*

*In the manuscript, the authors provided a comprehensive literature review on the backgrounds of their current study. The authors provided detailed information on methods for wind calculation and system calibration based on airborne platform (in Supplement), and gave evidence that their measured wind vector was insusceptible of lift-induced upwash and leverage effect. From these aspects, I think the authors have solved the difficulties on wind vector measurement from airborne platform very well. My major criticism is in the evaluation of UAV EC turbulent flux measurements. How they calculated the fluxes of sensible heat, latent heat, and CO2 from UAV are not clear stated. Can the results of error analysis results for turbulent fluxes measured by UAV EC system represent the actual situation, or whether the Monte Carlo simulation methods is appropriate for error analysis of EC flux. Therefore, I think this work needs some improvement before it can be published.*

Re: Thank you for your insightful comments. In the revised manuscript, we have substantially revised this manuscript in both the methodology for error assessment and the relative contents. In particular, aspects involving the calculation of turbulent fluxes (including the necessary corrections) and the error analysis of wind and flux measurements have been thoroughly revised. Your comments are very helpful to improve the quality of the manuscript.

**Specific Comments**

*Q1. The approach for calculating the sensible heat, latent heat, and CO2 fluxes, as well as the friction velocity from the airborne (or UAV) EC measurements needs to be described in Supplement or manuscript.*

**Re:** In accordance with your comment, in the revised Supplement Part B, we added a detailed explanation to describe the equations to calculate the fluxes of sensible heat, latent heat, carbon dioxide ($CO_2$), and the methods to quantify the measurement uncertainty in them due to instrument

noise.

***Q2. Figs. 1 and 2, the underlying surface should be added in the background of the figures. In the case of low-altitude flight observation, the underlying surface has a direct effect on the EC measurements.***

Re: In accordance with your comment, in the revised manuscript, the information of underlying surface over the region for conducting flight campaign was also provided in Figs. 1 and 2. We used Sentinel-2A satellite image to depict the information of underlying surface.

***Q3. Line 172, the abbreviations CST should be defined at the first use in the manuscript.***

Re: The ambiguous abbreviation "CST" was revised as follow:

Lines 186-187, in Section 2.2.1: "The calibration flight was executed between 7:28-7:48 a.m. (China Standard Time, CST) to coincide with the ebb tide stage."

***Q4. Lines 258-264, this sentence is difficult to follow and confused me. Calculated the accurate turbulent flux value is important, but the authors stated that the objective is not to quantify the actual flux value. The authors should reorganize the sentence to clearly state the objective of flux calculation or evaluation in this paper.***

Re: In the revised manuscript, the original sentence (Lines 258-264) was rewritten, and the contents about error analysis for measurement of wind and turbulent flux have been substantially revised. The original used of Monte Carlo error simulation method (Lines 273-280 in the original manuscript) to estimate the measurement error of geo-referenced wind and turbulent flux has been totally removed. Then, we used the partial derivatives of the full calculation equation for geo-referenced wind and turbulent flux to estimate the measurement error in wind and fluxes. Accordingly, two main revisions have been made as follow:

First, in the revised manuscript, we used the linearized Taylor series expansions derived by Enriquez and Friehe (1995) (in the revised Supplement Part A) to determine the sensitivities of each of the geo-referenced wind vector components with respect to the relevant variables. Then, combined these sensitivity terms to estimate the overall measurement error ($1\sigma$) in the geo-referenced 3D wind vector. The results were provided in Section 3.1 of the revised manuscript.

Second, we added a section (Section 2.4.2) to illustrate the methods for estimating flux measurement error caused by instrumental noise by combining the covariance uncertainty estimated by RS method (Eq. 6 in the revised manuscript) and the propagation of errors in flux correction terms (Eqs. S29-S31 in Supplement Part B). In this study, we mainly focused on the error caused by instrumental noise due to they are related not only to the system performance, but also to the minimum resolvable capability for the flux to be measured. The results were given in Section 3.2 of the revised manuscript.

*5) In the discussion, other factors (e.g., variation of the flight height, atmospheric conditions etc.) that were not considered in this study but have an impact on the reliability of the UAV EC measurements should be added or described.*

Re: In accordance with your comment, we added the description of other factors which influence the UAV EC measurement in the discussion.

Lines 687-690, in Discussion: "Lastly, it should be noted that the accuracy of the measured geo-referenced wind vector and turbulent flux from the UAV-based EC system is subject to the combination of many factors, mainly including sensor accuracy, UAV powerplant, UAV fluctuation (e.g., variation of the UAV attitude and flight height), and the atmospheric conditions during the measurements, etc."

*6) The limitations of airborne (or UAV) EC measurements should be summarized or mentioned.*

Re: In accordance with your comment, we added some summary of the limitations of airborne EC measurements in the Section of conclusions and further works (Section 5).

Lines 732-735, in Conclusions and further works: "Although UAV-based EC measurements have many advantages over manned aircraft and tower-based EC measurements, airborne EC measurements themselves have some shortcomings, such as flux measurement results hard to interpret (e.g., influence from surface heterogeneity, flux divergence, etc.), the measurements are restricted to short periods of time, and the interaction between the UAV and turbulence."

*7) The manuscript is overall clearly written, except some typos or very complex sentences (e.g. Line 583).*

Re: The language of this manuscript has been revised entirely, errors about grammar, spelling, punctuation, and phrasing have been corrected.

**References:**

Enriquez, A. G. and Friehe, C. A.: Effects of Wind Stress and Wind Stress Curl Variability on Coastal Upwelling, Journal of Physical Oceanography, 25, 1651-1671, https://doi.org/10.1175/1520-0485(1995)025<1651:EOWSAW>2.0.CO;2, 1995.

---

## Author Response (AR2)

**Response to comments from Referees**

We are truly grateful to reviewers' critical comments and thoughtful suggestions. In accordance with the comments from the reviewers, the manuscript has been thoroughly revised in content; the revisions have been marked in tracked changes. All references to figure(s), table(s), section(s), page(s), and line(s) refer to the revised manuscript unless otherwise stated.

First, in order to improve the readability, the whole manuscript has been moderately revised. There are four main revisions in the revised manuscript.

1) The title of the manuscript has been revised to "Quality Evaluation for Measurements of Wind Field and Turbulent Flux from a UAV-based Eddy Covariance System" for more clearly express the study purpose of this manuscript to readers.

2) The language of Introduction section was optimized to further clarify the scientific problems to be solved in this manuscript and the latest advances in related studies.

3) For logical consideration, there were some structural adjustments (e.g., lines 152-157) in the revised manuscript.

4) We also invited a native English speaker to improve the language and check the grammar, spelling, punctuation, and phrasing of the manuscript.

Second, in order to further highlight the objectives of this study, summarizing sentences of the developed UAV-based EC system have been added and some redundant sentences have been deleted in the revised manuscript.

1) In the Conclusions and further works section, summary of the advantages and disadvantage of this UAV-based EC system were added (lines 685-689).

2) To avoid ambiguity for the readers, some redundant sentences (e.g., lines 656-664 in the original manuscript) were deleted in the revised manuscript.

Third, the supplement material was also thoroughly revised carefully for easy understanding and reading to readers.

At last, the English grammars, acronyms, and various reference errors have been corrected. The reviewers gave a very detailed reviewing on this paper and these have really improved the paper's quality and readability. And the formats of the picture, table, and reference have been revised.

For each specific comment, we have made the detailed reply as follows.

**General Comments:**

*The aircraft-based EC flux method has been developed for turbulence measurements at regional scale, which bridging the scale gap between ground tower-based EC flux measurements and model-derived measurements. The manuscript develops an UAV-based EC flux system, and evaluated its precision performance and calibration methods. I think the work is valid and in general useful and appropriate for Atmospheric Measurement Techniques. I recommend that the*

*authors are required to do a moderate modification of the manuscript and figures before acceptance. They should ensure that manuscript and figures tell the same story and that more effort is put into making certain that dates and terminology are correct and consistent.*

**Re:** We are very grateful to your comments and efforts to improve the quality of this manuscript. In accordance with the comments from the reviewers, the manuscript has been thoroughly revised in contents and language. The title of this manuscript has been revised to "Quality Evaluation for Measurements of Wind Field and Turbulent Flux from a UAV-based Eddy Covariance System" for allowing the readers clearly get the main purpose of this study. The figures, tables, dates, and terminology in the manuscript were checked in detail. We also invited a native English speaker to improve the language and check the grammar, spelling, punctuation, and phrasing of the manuscript. The revisions have been marked in tracked changes. Thanks again for your kind work.

**Specific Comments**

*Q1. The knowledge gap of your scientific problems on the UAV-based EC flux methods to be solved in this study should further be clarified, and showed the state of the art of your scientific problems based on the literature review.*

**Re:** In order to clearly express the main objective to be solved in this manuscript, two improvements have been made in the revised manuscript:

1) The title of this manuscript is revised to "Quality Evaluation for Measurements of Wind Field and Turbulent Flux from a UAV-based Eddy Covariance System" for more clearly express the study purpose of this manuscript to readers.

2) The Introduction section of the of the manuscript has been moderately revised for further clarifying the scientific problems to be solved in this study, and the state of the art of scientific problems to be solved in this study have been updated as well.

*Q2. The components and configuration of the UAV-based EC flux system should be summarized and further show us the advantage and disadvantage of this UAV-based EC flux developed in this study.*

Re: The components and configuration of the developed UAV-based EC flux system were introduced and summarized in Section 2.1 of the revised manuscript. Detailed information on this UAV-based EC system was given by Sun et al. (2021). The advantages and disadvantage of this UAV-based EC system were summarized in the last section (Section 5 Conclusions and further works) in the revised manuscript, the text was modified as follow:

Lines 685-689, in Section 5 of the revised manuscript: "The UAV-based EC system has several advantages over manned aircraft, including less turbulence disturbance in wind measurement, lower measurement altitude (above the ground level), simpler operation, and lower operating costs, etc.

However, there are still some shortcomings need to be overcome, such as resonance noise, how large the difference compared to the tower-based EC under the same conditions, and how to interpret the instantaneous flux results for the flight scenario (e.g., influence from surface heterogeneity, flux divergence, etc.)."

Then based on the disadvantages of the current system, further improvements works are introduced (Lines 689-696).

Sun, Y., Ma, J., Sude, B., Lin, X., Shang, H., Geng, B., Diao, Z., Du, J., and Quan, Z.: A UAV-Based Eddy Covariance System for Measurement of Mass and Energy Exchange of the Ecosystem: Preliminary Results, Sensors, 21, 10.3390/s21020403, 2021.

***Q3. The writing logic and result representation should be further optimized, and all materials should be centered on your scientific problems to be solved.***

**Re:** The whole manuscript has been thoroughly revised in language, logic, and expression for easy understanding and reading to readers. For example, lines 167-172 in the original manuscript have been reorganized in the revised manuscript (lines 152-157) for better understanding. Some redundant and ambiguous sentences (e.g., lines 102-104, 656-664, etc. in the original manuscript) were deleted in the revised manuscript for emphasizing the focus of the scientific problems to be solved.

***Q4. The methodology of the UAV-based EC flux methods should be further clarified on how to achieve the three-dimentional velocity and direction, trace gas concentrations and turbulent fluxes.***

**Re:** The detailed methodology on how to calculating the 3D wind vector, turbulent fluxes, and their uncertainties were provided in the Supplement material. The gas concentrations of $CO_2$ and water vapor were directly measured by an open path infrared gas analyzer (IRGA) mounted on the UAV, which was also clarified in the revised manuscript (lines 149-150).

The supplement material of this manuscript was also improved in language and expression for providing the readers with a clearly understanding of the methodology used in this manuscript.